# Protease-activated receptor-2 ligands reveal orthosteric and allosteric mechanisms of receptor inhibition

Amanda J. Kennedy[1,12], Linda Sundström [1], Stefan Geschwindner[2], Eunice K. Y. Poon[3], Yuhong Jiang[3], Rongfeng Chen[4], Rob Cooke[5], Shawn Johnstone[6], Andrew Madin[7], Junxian Lim [3], Qingqi Liu[4], Rink-Jan Lohman[3], Anneli Nordqvist [8], Maria Fridén-Saxin[9], Wenzhen Yang[4], Dean G. Brown[10], David P. Fairlie[3] & Niek Dekker [11✉]

Protease-activated receptor-2 (PAR2) has been implicated in multiple pathophysiologies but drug discovery is challenging due to low small molecule tractability and a complex activation mechanism. Here we report the pharmacological profiling of a potent new agonist, suggested by molecular modelling to bind in the putative orthosteric site, and two novel PAR2 antagonists with distinctly different mechanisms of inhibition. We identify coupling between different PAR2 binding sites. One antagonist is a competitive inhibitor that binds to the orthosteric site, while a second antagonist is a negative allosteric modulator that binds at a remote site. The allosteric modulator shows probe dependence, more effectively inhibiting peptide than protease activation of PAR2 signalling. Importantly, both antagonists are active in vivo, inhibiting PAR2 agonist-induced acute paw inflammation in rats and preventing activation of mast cells and neutrophils. These results highlight two distinct mechanisms of inhibition that potentially could be targeted for future development of drugs that modulate PAR2.

[1] Mechanistic Biology & Profiling, Discovery Sciences, R&D, AstraZeneca, Gothenburg, Sweden. [2] Structure & Biophysics, Discovery Sciences, R&D, AstraZeneca, Gothenburg, Sweden. [3] Centre for Inflammation and Disease Research (CIDR) and ARC Centre of Excellence in Advanced Molecular Imaging, Institute for Molecular Bioscience, University of Queensland, Brisbane, QLD 4072, Australia. [4] Pharmaron Beijing Co., Ltd., 6 Taihe Road BDA, 100176 Beijing, People's Republic of China. [5] Sosei Heptares, Steinmetz Building, Granta Park, Great Abington, Cambridge CB21 6DG, UK. [6] Department of Chemistry, IntelliSyn Pharma, 7171 Frederick-Banting, Montreal, QC H4S 1Z9, Canada. [7] Hit Discovery, Discovery Sciences, R&D, AstraZeneca, Cambridge, UK. [8] Medicinal Chemistry, Cardiovascular, Renal & Metabolism (CVRM), BioPharmaceuticals R&D, AstraZeneca, Gothenburg, Sweden. [9] Alliance & Project Management, Clinical Pharmacology and Safety Sciences, R&D, AstraZeneca, Gothenburg, Sweden. [10] Hit Discovery, Discovery Sciences, R&D, AstraZeneca, Boston, MA, USA. [11] Discovery Biology, Discovery Sciences, R&D, AstraZeneca, Gothenburg, Sweden. [12] Present address: MSD at the Francis Crick Institute, London, UK. ✉email: niek.dekker@astrazeneca.com

Protease-activated receptors (PARs) are a unique family of G protein-coupled receptors with an unusual activation mechanism. Endogenously, the N terminus of these receptors is cleaved by extracellular proteases to reveal a tethered ligand sequence which can intramolecularly bind and activate the receptor[1,2]. Protease-activated receptor-2 (PAR2) is predominately activated by serine proteases, such as trypsin and tryptase, which cleave between $R^{35}$–$S^{36}$ thereby exposing a new N-terminal sequence SLIGKV–, that acts as a tethered agonist for intramolecular PAR2 activation[3–5]. The activated receptor stimulates multiple different G protein-dependent and independent signalling pathways. PAR2 has been shown to signal through Gα proteins ($G_{q/11}$, $G_s$ and $G_{12/13}$) across different cell environments, as well as through G protein-independent proteins including β-arrestins 1/2[3,6,7]. In addition, synthetic peptide agonists corresponding to the tethered ligand sequence, either human SLIGKV-$NH_2$ or the rodent sequence SLIGRL-$NH_2$, can activate these signalling pathways via PAR2[8–10].

PAR2 has been shown to have roles in pain and migraine[6,11], cancer[12,13], metabolic disease, both obesity[14] and cardiovascular[15,16], as well as in inflammation and inflammatory diseases[17,18]. The wide range of effects of PAR2 underscores the importance of this receptor in human physiology and disease. As a consequence, this target is highly sought after in drug discovery and for a number of years has been a focus of major pharmaceutical endeavours[19]. However, as yet the discovery of an effective drug has proved challenging for PAR2. The only marketed drug for a PAR is vorapaxar, a selective antagonist of PAR1 that is an antiplatelet treatment for improving restricted blood flow[20]. Even though PAR2 has close sequence homology with PAR1, the discovery of PAR2 antagonists has been less successful, with weak antagonists that only inhibit selected signalling pathways (e.g. GB88)[21,22], or that show agonist properties in some cell types (e.g. C391)[23]. Development of tool compounds is important for better understanding of the mechanisms of PAR2 activation on different cell and tissue types and in diseases where PAR2 is a key mediator and potential therapeutic target.

In 2017, we reported the crystal structures of PAR2 bound to AZ8838 and AZ3451[24]. AZ8838 was derived from an initial high-throughput screen hit[24] and AZ3451 was obtained from a DNA-encoded library technology screen, which also found an agonist, 'compound 1'[25]. AZ8838 binds in an occluded pocket made up of transmembrane helices (TM) 1–3, 7 and extracellular loop 2 (ECL2), whereas AZ3451 occupies a pocket that faces the lipid bilayer and is formed by TM 2, 3 and 4. However, the definition of the orthosteric site of the tethered ligand remains elusive due to the lack of an agonist-bound PAR2 crystal structure. More recently, with the AZ8838-bound structure as a starting point, we used an extensive combinatorial approach of site-directed mutagenesis and computational modelling to propose the putative orthosteric site[26].

Here, we present the pharmacological characterisation of a novel agonist of PAR2 (AZ2429) and two novel antagonists (AZ8838 and AZ3451). Agonist AZ2429 is proposed to bind at the same site in PAR2 as the activating peptide SLIGKV-$NH_2$ but is a more potent activator of multiple PAR2-dependent signalling pathways. Antagonists AZ8838 and AZ3451 can inhibit both G protein-dependent and independent pathways via PAR2 in vitro and exert anti-inflammatory effects in vivo in a rat model of PAR2 agonist-induced paw oedema. Our approach to interrogate ligand-receptor interactions, using molecular modelling and surface plasmon resonance in combination with functional assays, characterises AZ8838 as a competitive antagonist, whilst AZ3451 is a negative allosteric modulator. Our findings highlight coupling between ligand binding sites in the PAR2 receptor and illustrate opportunities for both orthosteric and allosteric inhibitors of PAR2 functions in vitro and in vivo.

## Results

**Two chemical series of PAR2 antagonists.** Using the stabilised (StaR) PAR2 receptor[24], two novel antagonist series were discovered (Fig. 1a). For the first series, the initial hit compound of the benzimidazole series, AZ8935, was identified in a DNA-encoded library technology screen[25]. Chemical expansion of this series was carried out to provide analogues with improved potency across different PAR2-mediated signalling pathways (Fig. 1a and Table 1). Substitution at the para-position of the phenyl ring with a nitrile group (AZ3451) led to the biggest increase in potency over the unsubstituted analogue AZ8935. Heterocycles with a heteroatom in the para-position (AZ2623) also had a gain in potency relative to AZ8935. Modification of the cyclohexyl ring to a tetrahydropyran (AZ7126) was tolerated with only a modest decrease in potency in $Ca^{2+}$ and IP1 (inositol-1-phosphate) signalling, relative to the most potent analogue, AZ3451. The imidazole series was developed from an initial weak high-throughput screen hit[24]. Optimisation resulted in compound AZ8838, although a constrained heterocycle (AZ0107) also maintained similar potency (Fig. 1a and Table 1).

**PAR2 antagonists inhibit multiple signalling pathways.** The binding affinities of each series were compared for competitive binding with a europium-tagged fluorescent derivative of the known PAR2 agonist 2f-LIGRLO-$NH_2$ to Chinese hamster ovary cells transfected with human PAR2 (CHO-hPAR2). All compounds tested bound to human PAR2 (Table 1 and Fig. 1b); the benzimidazoles (AZ3451 $pK_i = 6.9 \pm 0.2$) bound with higher affinity than the imidazoles (AZ8838 $pK_i = 5.2 \pm 0.1$). In agreement, both series bound to HEKexpi293F membranes expressing hPAR2 (AZ3451 $pK_i = 7.9 \pm 0.1$ and AZ8838 $pK_i = 6.4 \pm 0.1$) in competition binding assays against $^3$H-acetylated-GB110. Throughout this manuscript, the reference agonists GB110, 2f-LIGRLO-$NH_2$ and SLIGRL-$NH_2$ are interchanged as they have each been shown to bind at a common or overlapping site within PAR2 and are reported to activate the receptor in a similar manner[26,27].

Antagonists from the imidazole and benzimidazole series were profiled against peptide-induced activation of different PAR2-mediated signalling pathways (Table 1). Activation of PAR2 by peptide agonist SLIGRL-$NH_2$ resulted in stimulation of the $G_q$ pathway triggering the $Ca^{2+}$ mobilisation and inositol phosphate (IP1) signalling cascade. Inhibition of $Ca^{2+}$ flux and accumulation of IP1 was evaluated in 1321N1 cells stably expressing human PAR2 (1321N1-hPAR2) using FLIPR and IP1 HTRF assays respectively. The compounds were potent antagonists against SLIGRL-$NH_2$ in the $Ca^{2+}$ assay with the benzimidazole series (AZ3451 $pIC_{50} = 8.6 \pm 0.1$) more potent than the imidazole series (AZ8838 $pIC_{50} = 5.70 \pm 0.02$, Fig. 1c). None of the compounds showed agonist activity by activating PAR2 in the $Ca^{2+}$ assay. AZ3451 and AZ8838 showed a similar potency trend when inhibiting IP1 production ($pIC_{50} = 7.65 \pm 0.02$ and $5.84 \pm 0.02$, respectively, Fig. 1d). Besides the canonical G protein signalling, PAR2 is known to activate other signalling pathways that are relevant in different diseases. The ability of these compounds to block peptide-induced activation of additional pathways was evaluated using a β-arrestin-2 recruitment assay based on enzyme complementation and an HTRF-based detection of phosphorylation of ERK1/2 in U2OS cells stably expressing human PAR2 (U2OS-hPAR2) (Table 1). AZ3451 and AZ8838 attenuated both peptide-induced phosphorylation of ERK1/2 ($pIC_{50} = 6.44 \pm 0.03$ and $5.7 \pm 0.1$, respectively, Fig. 1e) and β-arrestin-2 recruitment ($pIC_{50} = 7.06 \pm 0.04$ and $6.1 \pm 0.1$, respectively, Fig. 1f). Compounds from both series retained their activity on rat PAR2; inhibition of peptide-induced $Ca^{2+}$ flux in a

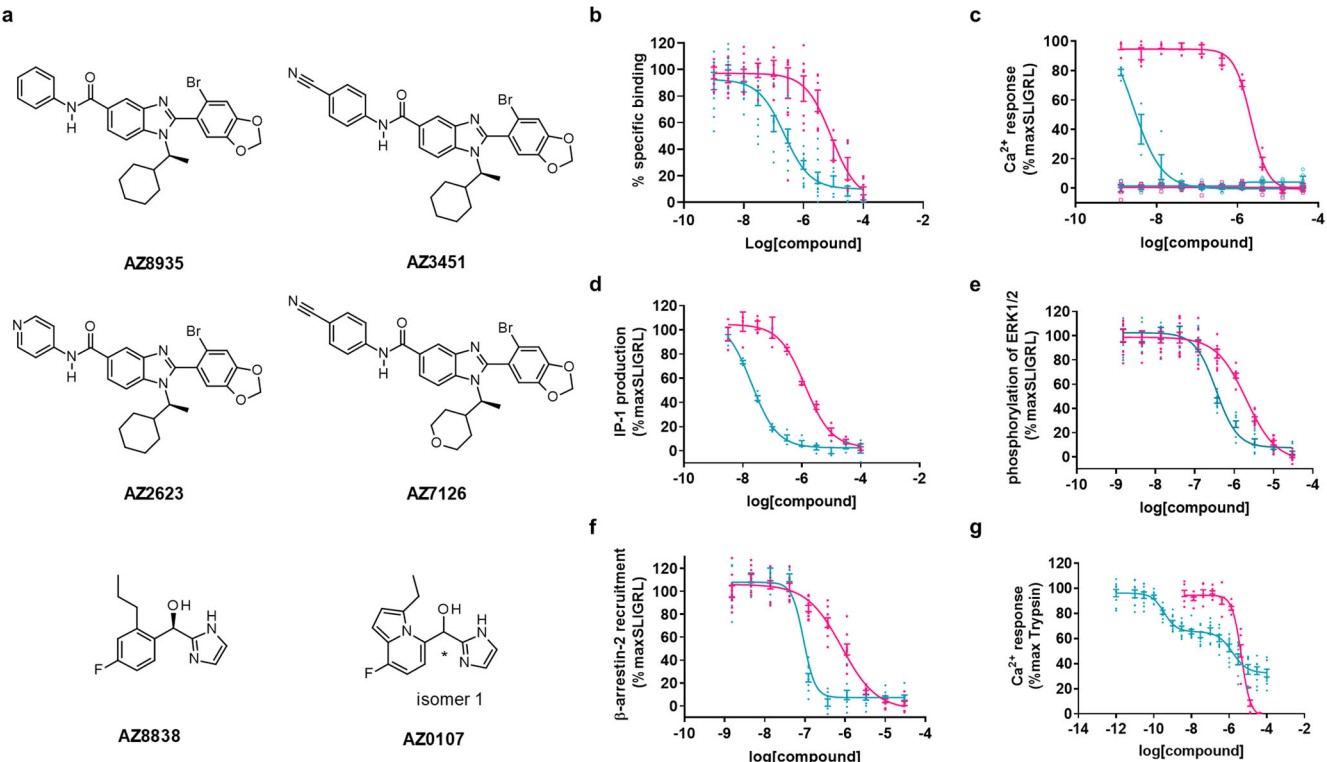

**Fig. 1 Pharmacological characterisation of PAR2 antagonists. a** Chemical structures of novel PAR2 antagonists. Benzimidazole antagonist (AZ8935) and optimised analogue structures (AZ3451, AZ2623 and AZ7126). Imidazole antagonist (AZ8838) and representative series analogue (AZ0107). **b** Binding of both series was evaluated in competition binding experiments against 2f-LIGRLO-(dtpa)Eu in CHO-hPAR2 cells. AZ8838 (magenta) and AZ3451 (cyan) had no effect when tested as agonists (open symbols) and inhibited SLIGRL-NH$_2$ (SLIGRL) activation (closed symbols) of **c** Ca$^{2+}$ mobilisation, **d** IP1 formation, both in 1321N1-hPAR2 cells and **e** phosphorylation of ERK1/2 and **f** β-arrestin-2 recruitment, both in U2OS-hPAR2 cells. **g** Both series also inhibited trypsin activation of PAR2 in Ca$^{2+}$ mobilisation in 1321N1-hPAR2 cells but interestingly AZ3451 gave a biphasic response whereas AZ8838 was monophasic. Inhibition of trypsin was evaluated in the presence of 1 μM Vorapaxar to suppress any contributions by PAR1 endogenously present in 1321N1 cells. Antagonist responses were calculated as % inhibition of agonist response in the absence of the antagonist. Graphs show representative data of 2 or more experiments, presented as individual data points with error bars denoting s.e.m. The comprehensive pharmacological data are stated in Table 1.

rat KNRK cell line with endogenous expression of rat PAR2 was confirmed (Table 1) prior to conducting in vivo experiments in the rat. The two distinct chemical series, benzimidazole and imidazoles, demonstrated potent inhibition across multiple PAR2-mediated signalling pathways, with the benzimidazole series consistently being more potent antagonists than the imidazole series.

The antagonists were also evaluated for their ability to block activation of PAR2 signalling by trypsin, which produces an endogenous tethered ligand in situ. For AZ8838, inhibition of trypsin-induced Ca$^{2+}$ mobilisation was sigmoidal and mono-phasic (pIC$_{50}$ = 5.40 ± 0.02, Fig. 1g), as observed for inhibition of peptide-induced Ca$^{2+}$ mobilisation (Fig. 1c and Supplementary Fig. 1). AZ8838 completely antagonised the agonist response at low micromolar concentrations. Similarly, antagonist AZ3451 displayed potent and complete monophasic inhibition of peptide-induced activation of PAR2 (Fig. 1c). However, when PAR2 was activated by trypsin, the response of AZ3451 was biphasic with partial inhibition at low nanomolar concentrations followed by some further inhibition at low micromolar concentrations (Fig. 1g). Careful analysis of the Ca$^{2+}$ response of all antagonists in the benzimidazole series revealed weak but reproducible suppression of signalling in a similar biphasic manner to that shown for AZ3451 (Supplementary Fig. 2). This biphasic response was also observed when tryptase was used instead of trypsin and was not due to inhibition of the enzyme

(Supplementary Fig. 3). In addition, it was not due to interference with the PAR1 receptor, as a similar response was seen in the presence of PAR1 antagonist Vorapaxar (Supplementary Fig. 4a, b); and was observed across a number of different cell backgrounds (Supplementary Fig. 4b–d). A similar partial inhibition of trypsin (but full inhibition of peptide) activation was apparent when monitoring phosphorylation of ERK1/2 in CHO-hPAR2 (Supplementary Fig. 4e, f). The crystal structure reveals that AZ3451 binds to a site facing the membrane with G/A157 and F154 highlighted as necessary for interaction[24]. Increasing the size of G157, by mutating to cysteine, or removing the side chain of the phenylalanine [F154A] disrupted the biphasic response showing only inhibition at low micromolar concentrations (Supplementary Fig. 5). These mutagenesis studies suggest that AZ3451 binds with high affinity to the site observed in the PAR2-AZ3451 crystal structure, but, intriguingly, has a second lower affinity binding site within the receptor which is only accessible when PAR2 is activated by the tethered ligand mechanism.

**AZ2429 is a novel potent agonist of PAR2 in vitro**. From DNA-encoded library screening, one agonist series was identified[25]. This structure carried three chiral centres and represented a mixture of four diastereomers. This mixture underwent chiral purification and three diastereomers were recovered in pure form. A single diastereomer AZ2429 (Fig. 2a) was identified as the most

**Table 1 In vitro pharmacology of PAR2 antagonists.**

| Compounds | $Ca^{2+}$ flux (hPAR2) | $Ca^{2+}$ flux (rPAR2) | IP1 (hPAR2) | pERK1/2 (hPAR2) | β-arrestin (hPAR2) | Competition Binding (hPAR2) |
|---|---|---|---|---|---|---|
| AZ8935 | 7.7 ± 0.1 ($n=3$) | 8.1 ± 0.1 ($n=4$) | 6.52 ± 0.04 ($n=6$) | 5.9 ± 0.1 ($n=3$) | 6.42 ± 0.03 ($n=3$) | nd |
| AZ3451 | 8.6 ± 0.1 ($n=30$) | 8.5 ± 0.1 ($n=12$) | 7.65 ± 0.02 ($n=122$) | 6.44 ± 0.03 ($n=3$) | 7.06 ± 0.04 ($n=3$) | 6.9 ± 0.2 ($n=4$) |
| AZ2623 | 8.3 ± 0.1 ($n=26$) | 8.3 ± 0.1 ($n=12$) | 7.09 ± 0.02 ($n=106$) | 6.1 ± 0.1 ($n=3$) | 6.31 ± 0.04 ($n=3$) | 7.1 ± 0.3 ($n=3$) |
| AZ7126 | 8.1 ± 0.2 ($n=4$) | 8.3 ± 0.2 ($n=4$) | 7.1 ± 0.1 ($n=6$) | 6.7 ± 0.1 ($n=3$) | 7.2 ± 0.1 ($n=3$) | nd |
| AZ8838 | 5.70 ± 0.02 ($n=33$) | 5.3 ± 0.1 ($n=15$) | 5.84 ± 0.02 ($n=158$) | 5.7 ± 0.1 ($n=3$) | 6.1 ± 0.1 ($n=3$) | 5.2 ± 0.1 ($n=3$) |
| AZ0107 | 6.1 ± 0.1 ($n=3$) | 5.5 ± 0.1 ($n=4$) | 6.7 ± 0.1 ($n=12$) | 5.9 ± 0.1 ($n=3$) | 6.3 ± 0.1 ($n=3$) | 5.4 ± 0.2 ($n=3$) |

In functional assays, PAR2 was activated with peptide, SLIGRL-NH2 in 1321N1-hPAR2 cells ($Ca^{2+}$ and IP1) or U2OS-hPAR2 (pERK1/2 and β-arrestin-2). Species differences were considered through monitoring $Ca^{2+}$ release by rat PAR2 in KNRK cells. Binding was measured in competition experiments against Eu-tagged-2f-LIGRLO(dtpa)-NH2 on CHO-hPAR2 cells. Data are presented as mean pKi (binding only) or pIC50 ± s.e.m. based on $n$ independent experiments.
*nd* Not determined.

potent agonist with activity at submicromolar concentrations (pEC50 = 6.70 ± 0.03) in an IP1 assay; the other two diastereomers were less potent (pEC50 = 4.99 ± 0.09, 5.45 ± 0.01).

Binding of AZ2429 to PAR2 was measured on HEKexpi293F membranes, expressing human PAR2, by displacement of [³H]-GB110. AZ2429 was able to displace the probe (p$K_i$ = 7.2 ± 0.2, Fig. 2b and Table 2) in a similar manner to that of unlabelled GB110 and SLIGKV-NH2. AZ2429 was a potent agonist of multiple signalling pathways at PAR2 (Fig. 2 and Table 2). AZ2429 activated PAR2-induced $Ca^{2+}$ signalling (pEC50 = 6.78 ± 0.03) with a 7-fold higher potency than SLIGKV-NH2 (pEC50 = 5.9 ± 0.1) and a 6-fold lower potency than peptide mimetic GB110 (pEC50 = 7.5 ± 0.1) (Fig. 2c). No $Ca^{2+}$ mobilisation could be detected in parental 1321N1 cells consistent with the observed FLIPR activity being specific for PAR2. When monitoring IP1 accumulation, AZ2429 had a similar potency to GB110 (pEC50 = 6.70 ± 0.03 and pEC50 = 6.5 ± 0.1, respectively) and was 120-fold more potent than SLIGKV-NH2 (pEC50 = 4.6 ± 0.1, Fig. 2d). AZ2429 stimulated β-arrestin-2 recruitment to PAR2 (pEC50 = 6.6 ± 0.1, Fig. 2e). No AZ2429 activity was detected against related receptors PAR1 and PAR4 in a β-arrestin-2 recruitment assay, supporting high selectivity for PAR2 (DiscoverX profiling service, Supplementary Table S1). AZ2429 also induced phosphorylation of ERK1/2 (pEC50 = 7.4 ± 0.1, Fig. 2f). Consistent with the data for activation of the $G_q$ pathway, AZ2429 was more potent than SLIGKV-NH2 in β-arrestin-2 recruitment and pERK1/2 assays (Table 2). AZ2429 was a full agonist with similar efficacy to the peptide SLIGKV-NH2, corresponding to the endogenous human tethered ligand sequence, but was more potent than SLIGKV-NH2 in all signalling pathways tested.

To investigate where the novel agonist AZ2429 binds in PAR2, molecular docking simulations were performed using the AZ8838-bound crystal structure of PAR2. Modelling placed the agonist AZ2429 in the previously reported binding site for SLIGKV-NH2 in PAR2[26]. The underlying assumption was that AZ2429 and the DNA-encoded compound (DNA-AZ2429) bind to the receptor in a similar way. Docking calculations were therefore guided to bury the benzyltriazole deep into the receptor pocket leaving the methylamide facing the extracellular region to allow for the attachment of the DNA-tag, which was present when this compound was identified[25].

Docking predicts that the heterocycle of AZ2429 adopts a T-shape orientation towards H227, between polar residues D228 and Y82. This is the part of the binding pocket predicted to bind the N-terminal serine of the tethered ligand and which binds the imidazole of AZ8838 in the crystal structure, although the triazole of AZ2429 is slightly closer to D228 (Fig. 2g). The benzyl group of AZ2429 extends into the AZ8838 binding pocket and forms a T-shape pi-interaction with F155 in a similar manner to the phenyl group of AZ8838 (Fig. 2g) and is placed 4.1 Å above the cation of K131. In agreement with the positioning of the peptide corresponding to the endogenous tethered ligand, the

cyclohexylamine side change fits into the hydrophobic pocket, surrounded by H310, Y232 and Y326, mimicking the interactions of the leucine side chain of SLIGKV-NH2. Towards the extracellular domain, the aminotetralin–carboxamide is positioned around the proposed binding region of the third hydrophobic residue of SLIGKV-NH2 fulfilling the prerequisite that the terminal amide of AZ2429 is freely accessible from the opening of the receptor pocket.

**Antagonists AZ8838 and AZ3451 have distinct modes of action.** Having identified the different binding sites (Fig. 3a), further characterisation of AZ8838, AZ3451 and AZ2429 compound binding was carried out using surface plasmon resonance (SPR) measurements. Binding characterisation of AZ8838 with SPR was previously reported[24], revealing slow binding kinetics and a corresponding residence time of about 84 min. In contrast, binding of the antagonist AZ3451 was characterised by high affinity with accompanying rapid association kinetics to the receptor and a residence time of about 2 min (Fig. 3b and Table 3). AZ2429 association kinetics were around 100-fold slower, whilst the dissociation rate only decreased 5-fold, resulting in a 10-fold weaker affinity (Fig. 3e and Table 3).

To investigate the interplay between the different ligand binding sites of PAR2, binding of AZ3451 and AZ2429 were first profiled, using single-cycle kinetics, in the presence of saturating amounts of AZ8838 in the running buffer. AZ3451 was able to bind to the receptor, albeit with a significant 3-fold reduction in affinity due to a decreased association rate in combination with an increased dissociation rate (Fig. 3d and Table 3), whereas binding of AZ2429 was completely abolished within the tested concentration regime (Fig. 3g). Those findings suggest that there is a long-range allosteric coupling between the AZ8838 binding site and the AZ3451 binding site on the outside of the helix bundle. The complete lack of agonist binding to the AZ8838-bound receptor is indicative of a physical hindrance impairing a constructive agonist binding process. Next, the allosteric coupling between the binding site of agonist AZ2429 and the binding site of antagonist AZ3451 was probed. AZ3451 was able to bind in the presence of AZ2429, and vice versa, but both showed significant alterations in their binding kinetics (Fig. 3c, f and Table 3). AZ3451 displays a significantly increased association rate to the AZ2429-bound receptor, that was entirely compensated by a more rapid dissociation from the receptor leaving the affinity essentially unchanged (Table 3). Interestingly, differences in binding kinetics were also observed for the reverse situation of AZ2429 binding to the AZ3451-bound receptor but showing the inverse behaviour. AZ2429 shows a significantly reduced association rate that is partly compensated by a significant increase in the residence time, leading to a small but significant reduction in the affinity (Table 3). Both findings are supporting each other and suggest, that there is a long-range

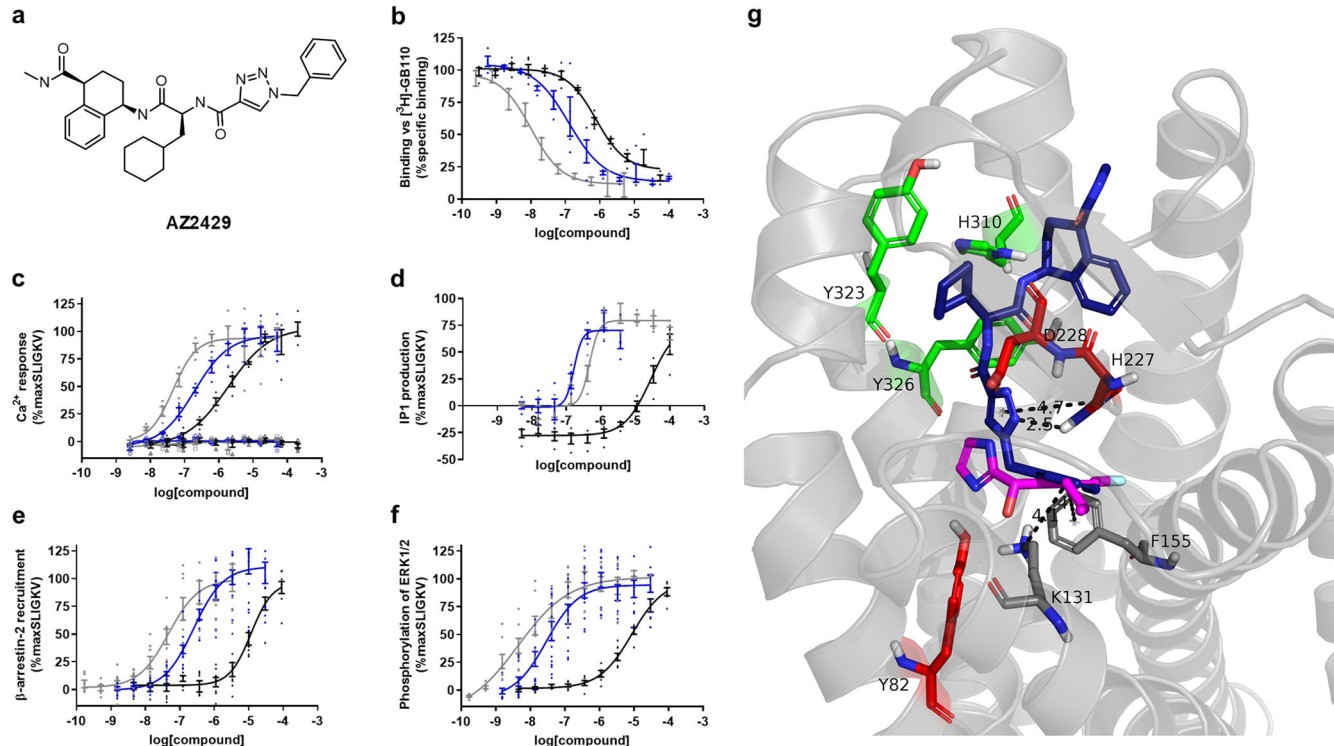

**Fig. 2 AZ2429 binds to PAR2 and activates multiple signalling pathways. a** Chemical structure of most active enantiomer of the potent agonist compound (AZ2429). AZ2429 (blue) and known PAR2 agonists SLIGKV-NH$_2$ (black) and GB110 (grey) were pharmacologically profiled for: **b** competitive binding against [$^3$H]-GB110 in membrane preparations of HEKexpi293 cells overexpressing human PAR2; **c** Ca$^{2+}$ mobilisation and **d** IP1 production in 1321N1-hPAR2 cells; as well as **e** β-arrestin-2 recruitment and **f** phosphorylation of ERK1/2 in U2OS-hPAR2 cells. No agonistic effects were detected in parental 1321N1 cells, exemplified in the Ca$^{2+}$ assay (open symbols). Agonist responses were calculated as % maximum response of SLIGKV-NH$_2$. Graphs show representative data of 2 or more experiments, presented as individual data points with error bars denoting the s.e.m. The comprehensive pharmacological data are stated in Table 2. **g** Refined model of AZ2429 (blue) in the PAR2 receptor generated through molecular docking calculations in an adapted model of the PAR2 structure. Superimposed is AZ8838 (magenta) from the crystal structure (PDB 5NDD) showing the overlap of the benzyl substituents of each compound. Important receptor side chains are colour-coordinated based on their spatial localisation.

---

**Table 2 In vitro pharmacology of agonist AZ2429 compared with known PAR2 agonists GB110 and SLIGKV-NH$_2$.**

| Compounds | Competition binding | Ca$^{2+}$ flux | IP1 | β-arrestin | pERK1/2 |
|---|---|---|---|---|---|
| AZ2429 | 7.2 ± 0.2 ($n = 4$) | 6.78 ± 0.04 ($n = 3$) | 6.70 ± 0.03 ($n = 6$) | 6.6 ± 0.1 ($n = 4$) | 7.4 ± 0.1 ($n = 4$) |
| GB110 | 8.2 ± 0.3 ($n = 2$) | 7.5 ± 0.1 ($n = 3$) | 6.5 ± 0.1 ($n = 8$) | 7.3 ± 0.1 ($n = 4$) | 8.2 ± 0.1 ($n = 4$) |
| SLIGKV-NH$_2$ | 6.33 ± 0.04 ($n = 5$) | 5.9 ± 0.1 ($n = 3$) | 4.6 ± 0.1 ($n = 6$) | 4.9 ± 0.1 ($n = 4$) | 5.0 ± 0.1 ($n = 3$) |

IP1 and Ca$^{2+}$ were assessed in 1321N1-hPAR2 cells, β-arrestin-2 and pERK1/2 in U2OS-hPAR2, binding was measured in membranes from HEKexpi293F cells. Data are presented as mean pKi (binding only) or pEC$_{50}$ ± s.e.m. based on $n$ independent experiments.

---

allosteric coupling between the AZ3451 and AZ2429 binding sites.

The interaction of binding sites within PAR2 was next probed pharmacologically in cells using Schild plot analysis of antagonist function in the presence of either a peptide or endogenous tethered ligand agonist. Increasing concentrations of AZ8838 caused a rightward shift in the response of 2f-LIGRL-NH$_2$ without a decrease of the maximum, giving a Schild slope = 1.1 (Fig. 4a). A similar response, with a Schild slope = 0.9, was observed in the presence of trypsin (Fig. 4b), consistent with AZ8838 being a competitive antagonist at the orthosteric site occupied by the tethered ligand. In both cases, the calculated Schild slope of AZ8838 was not statistically distinguishable from a hypothetical Schild slope of 1. In contrast, AZ3451 had saturable antagonist effects against both 2f-LIGRL-NH$_2$ and trypsin (Fig. 4c,

d), with Schild slopes of 0.4 and 0.3, respectively; consistent with it being a non-competitive antagonist. Analysing the inhibition by AZ3451 using an allosteric model (Supplementary Fig. 6a, b) based on Ca$^{2+}$ mobilisation allowed calculation of $\alpha$ and $\beta$ values for 2f-LIGRL-NH$_2$, log$\alpha = -1.98 \pm 0.11$; log$\beta = -0.28 \pm 0.07$ and trypsin, log$\alpha = -1.19 \pm 0.07$; log$\beta = -0.66 \pm 0.08$. A second model of allosterism based on Ca$^{2+}$ mobilisation allowed calculation of relative activity and estimation of the $\gamma$ values for 2f-LIGRL-NH$_2$ and trypsin (Supplementary Table S2). The relative activity estimates the value of $\gamma$ to be 0.002 and 0.09 for 2f-LIGRL-NH$_2$ and trypsin respectively, indicating that AZ3451 caused a decrease in both affinity and efficacy (Supplementary Fig. 6c, d); consistent with a previous study showing that AZ3451 is a negative allosteric modulator of PAR2[26].

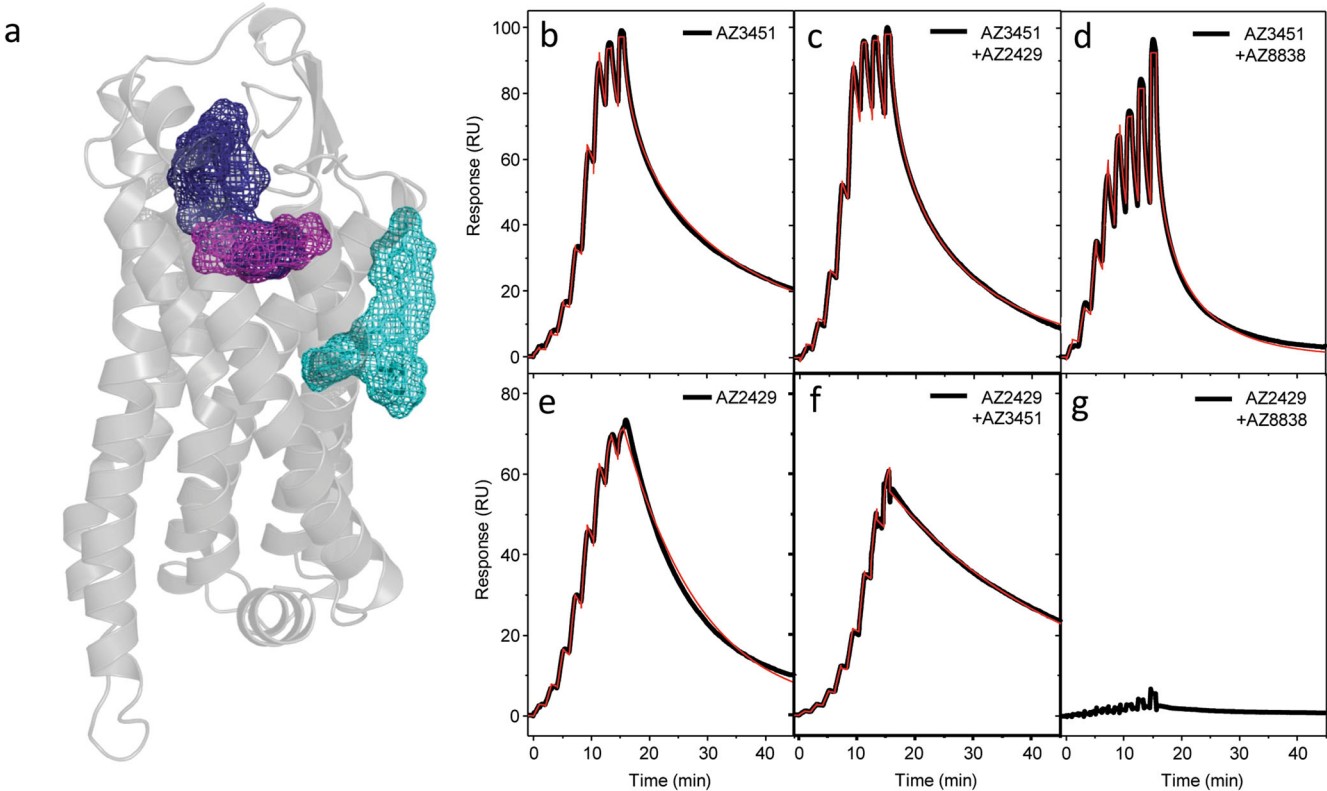

**Fig. 3 Interplay between antagonist and agonist binding sites of PAR2 analysed by SPR. a** Schematic representation of compound binding sites within PAR2: agonist AZ2429 binding site (blue) shows a partial overlap with the fragment binding site of AZ8838 (magenta), while the AZ3451 binding site (cyan) is on the outside of the helical bundle. Single-cycle kinetic measurements were used to generate the sensorgram profiles of AZ3451 binding to **b** apo-PAR2, **c** the PAR2-AZ2429 complex and **d** the PAR2-AZ8838 complex. AZ2429 binding to **e** apo-PAR2, **f** the PAR2-AZ3451 complex and **g** the PAR2-AZ8838 complex was measured in a similar manner. Black curves depict the raw data, whilst the red curves represent the fitting of the raw data to a 1:1 binding model. The extracted kinetic data are reported in Table 3.

---

**Table 3 Competitive binding kinetics of agonist and antagonists for PAR2 receptor quantified by SPR.**

| Compound | $k_a$ (M$^{-1}$s$^{-1}$) | Residence time (s) | p$K_D$ |
|---|---|---|---|
| AZ3451 | $1.1 \pm 0.6 \times 10^6$ ($n = 4$) | $116 \pm 68$ ($n = 4$) | $7.96 \pm 0.04$ ($n = 4$) |
| AZ3451 + 10 µM AZ2429 | $2.3 \pm 0.8 \times 10^6$++ ($n = 8$) | $47 \pm 15$+ ($n = 8$) | $7.96 \pm 0.04$ ($n = 8$) |
| AZ3451 + 50 µM AZ8838 | $6.7 \pm 1.1 \times 10^5$ ($n = 4$) | $44 \pm 7$ ($n = 4$) | $7.47 \pm 0.03$++++ ($n = 4$) |
| AZ2429 | $1.5 \pm 0.1 \times 10^4$ ($n = 4$) | $523 \pm 50$ ($n = 4$) | $6.89 \pm 0.03$ ($n = 4$) |
| AZ2429 + 1 µM AZ3451 | $3.7 \pm 0.4 \times 10^3$**** ($n = 8$) | $1719 \pm 100$**** ($n = 8$) | $6.80 \pm 0.04$** ($n = 8$) |
| AZ2429 + 50 µM AZ8838 | No binding ($n = 4$) | No binding ($n = 4$) | No binding ($n = 4$) |

The kinetic rate constants and affinities have been extracted from the data through fitting to a simple 1:1 single-cycle kinetic titration model as described under methods. Data are presented as mean values ± s.e.m. based on $n$ independent experiments. Statistical analysis has been performed by the Student's $t$-test or one-way ANOVA.
+$p < 0.05$, ++$p < 0.01$, +++$p < 0.001$, ++++$p < 0.0001$ compared to AZ3451.
*$p < 0.05$, **$p < 0.01$, ***$p < 0.001$, ****$p < 0.0001$ compared to AZ2429.

---

**Antagonists AZ3451 and AZ8838 are anti-inflammatory in vivo.** A single intraplantar injection of the potent and selective PAR2 agonist 2f-LIGRLO-NH$_2$ (350 µg per paw in saline) into the hind paws of Wistar rats caused an acute oedema manifested by paw swelling that peaked after 30 min and subsided within 4 h. PAR2 antagonists, AZ3451 (10 mg kg$^{-1}$ s.c. 30 min prior) and AZ8838 (10 mg kg$^{-1}$ p.o. 2 h prior) were evaluated for their capacity to inhibit this agonist-induced paw inflammation when administered prior to agonist and both showed 60% reduction of paw swelling at the doses given (Fig. 5a). Histological analyses of the rat paw tissue after agonist administration indicated that maximal paw inflammation after 30 min coincided with activation and degranulation of mast cells and neutrophils.

The agonist 2f-LIGRLO-NH$_2$ did not cause any changes in ED1-positive macrophages (Supplementary Fig. 7). Paw tissues showed increased numbers of activated mast cells (Fig. 5b, d), decreased tryptase-positive and intact/inactive mast cells (Fig. 5c, e), and increased histamine release (Fig. 5f) compared to saline-treated animals. This is consistent with agonist-induced degranulation of mast cells. Moreover, 2f-LIGRLO-NH$_2$ also caused degranulation of neutrophils in the paw as shown by a decreased number of neutrophil elastase-positive cells (Fig. 5g, h), with increased myeloperoxidase (MPO) activity (Fig. 5i). Pre-treatment of the rats with either of the two PAR2 antagonists was able to strongly reduce rat paw inflammation (Fig. 5a) by inhibition of both mast cell (Fig. 5b–f) and neutrophil (Fig. 5g–i) activation in the tissues.

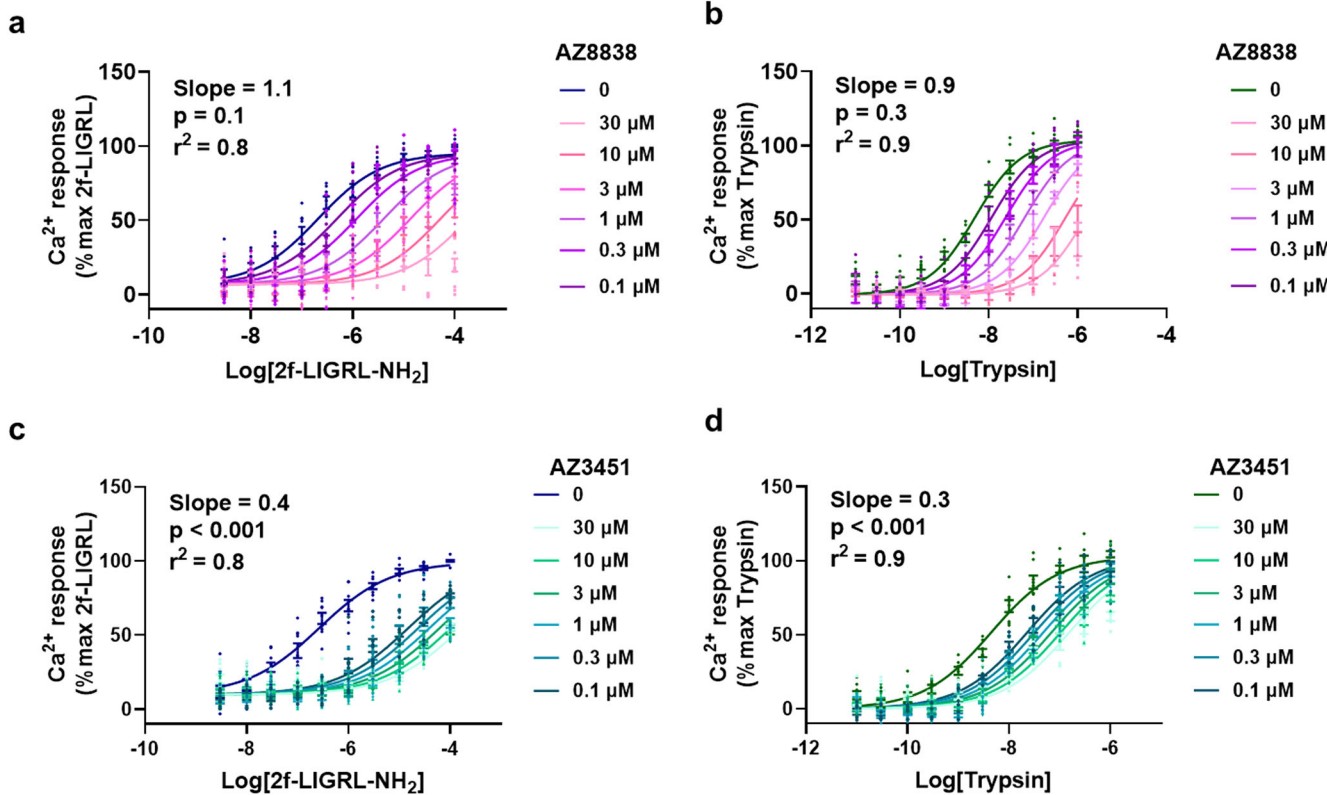

**Fig. 4 Mechanistic understanding of PAR2 antagonists.** Concentration–response curves for PAR2 agonists, 2f-LIGRL-NH$_2$ or trypsin, were measured in the presence of increasing concentrations of the two antagonists AZ8838 and AZ3451 in CHO-hPAR2 cells. AZ8838 was a competitive antagonist against **a** 2f-LIGRL-NH$_2$ ($n = 4$) and **b** trypsin ($n = 3$). In contrast, AZ3451 was a non-competitive antagonist against **c** 2f-LIGRL-NH$_2$ ($n = 3$) and **d** trypsin ($n = 3$). Data were analysed using Gaddum/Schild EC$_{50}$ global fitting competitive model and the calculated Schild slopes are shown. By comparing the calculated Schild slope of AZ8838 and AZ3451 versus a hypothetical Schild slope = 1 using extra-sum of squares $F$-test, the $p$-values for AZ8838 against 2f-LIGRL-NH$_2$ and trypsin are >0.05 indicating that the calculated Schild slope for AZ8838 is not statistically distinguishable from a hypothetical Schild slope = 1. While the $p$-values for AZ3451 against 2f-LIGRL-NH$_2$ and trypsin are <0.001 indicating that the calculated Schild slope for AZ3451 is statistically distinguishable from a hypothetical Schild slope = 1. Data are presented as individual data points with error bar denoting s.e.m. of $n > 3$ independent experiments.

## Discussion

Previous studies illustrate the challenges associated with the development of PAR2 antagonists. High millimolar concentrations of N1-3-methylbutyryl-N4-6-aminohexanoyl-piperazine (ENMD-1068) were required to dose-dependently attenuate joint inflammation[28]. The small molecule, GB88[29,30], at low micromolar concentrations, is a selective antagonist of PAR2-mediated Ca$^{2+}$ release in vitro and of rodent models of PAR2-induced paw odeoma[21], collagen-induced arthritis[21], TNBS-induced colitis[31] and diet-induced obesity[32]. However, GB88 shows biased signalling, blocking only the $G_q$ pathway in many[30], but not all[23], cell types, whilst being an agonist of cAMP, ERK1/2 and RhoA pathways in human cells[22]. C391 is a peptidomimetic antagonist derived from the 2f-LIGRLO-NH$_2$ sequence and also shows biased signalling, blocking the Ca$^{2+}$ and MAPK pathways in vitro at micromolar concentrations but stimulating Ca$^{2+}$ release at higher concentrations[23]. More recently, I-191 was identified as an inhibitor of PAR2 agonist-induced G protein-dependent signalling pathways in vitro[12], but its properties in vivo have not been reported. Herein we present two novel PAR2 antagonist series. AZ8838 was the optimised structure of the imidazole series and is an antagonist at low micromolar concentrations, of peptide- or protease-induced PAR2-mediated $G_q$, ERK1/2 and β-arrestin-2 signalling. It shows no agonist activity in the signalling pathways investigated and is anti-inflammatory in vivo after oral delivery

(or subcutaneous injection, Supplementary Fig. 8a) in a model of rat paw oedema induced by peptide agonists of PAR2. The optimised structure of the benzimidazole series, AZ3451, is a potent negative allosteric modulator at nanomolar concentrations of peptide-activated PAR2-mediated $G_q$, ERK1/2 and β-arrestin-2 signalling. It also shows no agonist activity in any of the signalling pathways tested and was active in vivo in rats after subcutaneous injection but not after oral delivery (Supplementary Fig. 8b). The identification of PAR2 antagonists that inhibit G protein-dependent and -independent signalling pathways in vitro and show anti-inflammatory activity in vivo in rodents provides two new tool compounds for interrogating PAR2 functions.

AZ2429, a novel agonist, discovered through a DNA-encoded library screen and identified after chiral separation, activates G protein-dependent and -independent signalling pathways via PAR2. With a similar potency to peptidomimetic GB110, this small molecule is more potent than the SLIGKV-NH$_2$ peptide agonist. Molecular modelling predicted that AZ2429 binds in the same pocket as SLIGKV-NH$_2$ [26].

Remarkably, unlike other PAR2 agonists, AZ2429 can bind to the PAR2 stabilised receptor (StaR), which contains several mutations to improve its thermostability. The mutations lock the receptor in an inactive-like configuration, closing off the agonist binding site such that the peptide/tethered ligand agonists are unable to bind[26]. Interestingly, agonist AZ2429 was able to bind,

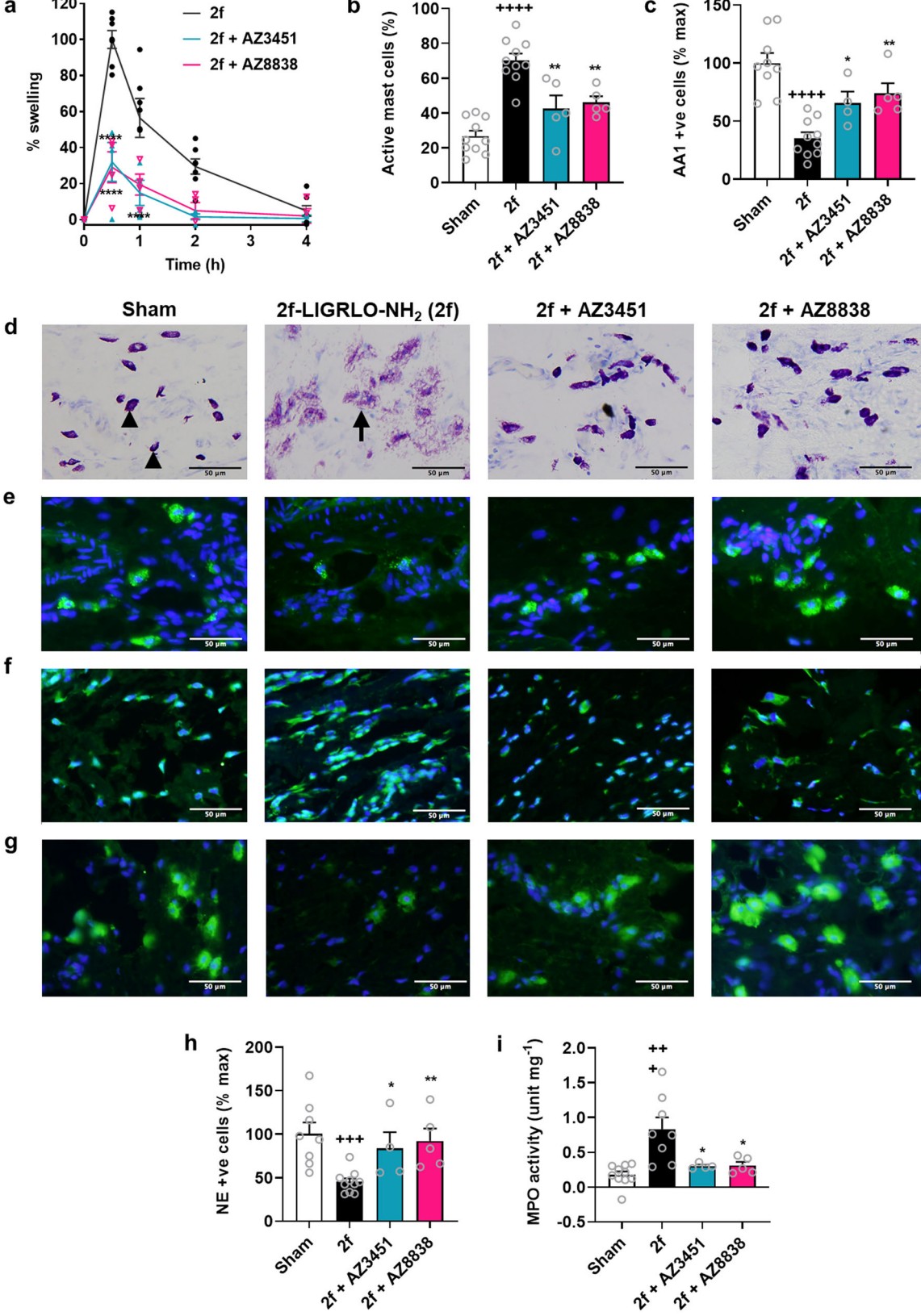

thought to be because of its small molecule nature and due to its heterocycle binding in a stable T-shape orientation between H227, D228 and Y82. This allowed for the first biophysical characterisation of interactions between the two distinct antagonist binding sites (AZ8838 and AZ3451) and the agonist binding site of PAR2. With this new tool, and supported by pharmacological characterisation in vitro, we have identified two distinct functional mechanisms for inhibition of PAR2.

**Fig. 5 AZ3451 and AZ8838 are anti-inflammatory in a PAR2 agonist-induced rat paw oedema model. a** PAR2 agonist 2f-LIGRLO-NH$_2$ (2 f, 350 μg/paw, i.pl.) caused peak paw swelling at 30 min. AZ3451 (10 mg kg$^{-1}$ s.c.) 30 min before, or AZ8838 (10 mg kg$^{-1}$ p.o.) 2 h before, agonist 2f-LIGRLO-NH$_2$ significantly reduced paw swelling. All subsequent measurements are at 30 min after i. pl. saline (sham) or 2f-LIGRLO-NH$_2$ ( ± antagonist) groups. **b** Percentage activated/degranulated mast cells in paw induced by 2f-LIGRLO-NH$_2$ was inhibited by AZ3451 or AZ8838 pre-treatment. **c** Tryptase-positive (AA1+ve) intact mast cells in paw (% cells in sham) were decreased by 2f-LIGRLO-NH$_2$, which was inhibited by AZ3451 or AZ8838 pre-treatment. **d** Intact mast cells stained dark purple by toluidine blue (black arrowheads) and activated/degranulated mast cells stained light purple (black arrow). **e** Tryptase-positive mast cells (green) were significantly reduced by 2f-LIGRLO-NH$_2$ vs sham; this was partly blocked by AZ3451 or AZ8838 pre-treatment. Extracellular tryptase undetectable by IHC. **f** Histamine release (green) was blocked by AZ3451 or AZ8838 pre-treatment. Sham showed intact/inactive histamine-containing mast cells, whereas 2f-LIGRLO-NH$_2$ showed extracellular histamine indicating mast cell degranulation. AZ3451- and AZ8838-treated paws contained inactive mast cells with minimal histamine. **g** Neutrophil elastase-positive (NE+ve) intact neutrophils (green) in paw. 2f-LIGRLO-NH$_2$ decreased NE+ve cells, which was partially inhibited by AZ3451 or AZ8848 pre-treatment. **h** % NE+ve cells vs sham in paw at 30 min after 2f-LIGRLO-NH$_2$ or saline. **i** Myeloperoxidase (MPO) activity, indicating neutrophil activation, was increased by 2f-LIGRLO-NH$_2$ but reduced almost to baseline by pre-treatment with AZ3451 or AZ8838. IHC images: nuclei stained blue with DAPI. Data from two independent experiments shown as mean ± s.e.m., $n = 10$ for sham and 2f-LIGRLO-NH$_2$ control, $n = 5$ for AZ8838 and AZ3451 groups. Analysis: Student's $t$-test or one-way ANOVA; $^+p < 0.05$, $^{++}p < 0.01$, $^{+++}p < 0.001$, $^{++++}p < 0.0001$ compared to sham; $*p < 0.05$, $**p < 0.01$, $***p < 0.001$, $****p < 0.0001$ compared to 2f-LIGRLO-NH$_2$ control.

In SPR studies, when AZ8838 was bound to the receptor, AZ2429 was no longer able to bind, consistent with overlapping binding sites. The predicted docking pose of AZ2429 agrees with the SPR data and visualises how AZ8838 may bind in a subpocket of the agonist binding site. We show that AZ8838 is a competitive antagonist of 2f-LIGRL-NH$_2$ building on a previous study[26] which indicated key receptor residues necessary for peptide-induced receptor activation. Importantly, we confirm that AZ8838 is also a competitive antagonist of the endogenous tethered ligand, suggesting that it may bind at the same site in PAR2 as the tethered ligand.

In contrast, SPR studies identified allosteric coupling between the more remote binding sites of AZ3451 and AZ2429. AZ2429 had altered binding kinetics in the presence of AZ3451 and vice versa. The agonist may help stabilise intermediate active conformation(s) of PAR2 that are distinct from the ground state of the receptor, which could lead to the observed reduction in residence time and thus decrease in binding for the antagonist. Like other extrahelical antagonists[33], AZ3451 could restrict inter-helical conformational rearrangements. Thal et al. suggest that upon binding to its site between TM 2 and 4, AZ3451 could prevent the rotation and movement of TM3 required for receptor activation. This would render the receptor in a more stable ground state conformation which could explain the significant increase in residence time of AZ2429. Pharmacological analysis of the antagonist activity of AZ3451, in the presence of peptide or protease agonists of PAR2, supports the SPR binding studies; indicating that this antagonist is non-competitive (Schild slope ≪ 1.0) and a negative allosteric modulator ($\alpha < 1$). The data are consistent with it causing reduced agonist binding to PAR2 and the decreased association rate of binding of AZ2429 in the presence of AZ3451.

Moreover, SPR studies showed that both antagonists could bind simultaneously to PAR2, as would be expected with the well-separated binding sites revealed in the protein crystal structures. The almost unchanged binding kinetics of AZ3451 in the presence of AZ8838 suggest that their binding sites are not conformationally linked, although it is noteworthy that the affinity is reduced.

By combining molecular modelling, SPR and in vitro pharmacological characterisation, we present a newfound understanding of the interaction between binding sites in PAR2. At the putative orthosteric site, located between TM1, 3, 6, 7 and ECL2, binding of the tethered ligand, peptide or small molecule (AZ2429) agonists causes receptor activation. AZ8838 appears to bind at a site overlapping this orthosteric site to inhibit receptor signalling in a competitive manner. AZ3451 binds to a more

remote site on the outside of the 7TM helical bundle and inhibits receptor signalling from this allosteric site through bidirectional, long-range coupling to the orthosteric site.

Allosteric antagonist, AZ3451 displayed intriguing pharmacology. In binding experiments against labelled peptidomimetic agonists, there was monophasic competition, and this was consistent with the inhibition of peptide-induced activation of PAR2 signalling. However, AZ3451 inhibition of the activation of PAR2 signalling by trypsin was biphasic (pIC$_{50}$ = 8.5 and 6), despite it being well-characterised that protease activation of PAR2 is monophasic[4,6,34]. Therefore, AZ3451 exhibits probe dependence, an intriguing characteristic of allosteric antagonists[35], which has previously been characterised on muscarinic receptor subtypes[36]. The biphasic effect was independent of assay format, cell line and receptor expression level; being observed in three different cell lines, 1321N1-hPAR2 and CHO-hPAR2 stably expressing human PAR2 and HT-29 cells with endogenous levels of PAR2, and two different signalling pathways (Ca$^{2+}$ and ERK1/2). The biphasic response was not perturbed by a PAR1 antagonist, suggesting it was not due to interaction with PAR1. Mutagenesis around the known AZ3451 binding site appeared to disrupt the biphasic response. AZ3451 inhibition of trypsin-induced PAR2 activation at mutant receptors [F154A and G157C] was monophasic with a lower affinity (pIC$_{50}$ = 6), suggesting that AZ3451 may have a second lower affinity binding site in PAR2. Although a second binding site for AZ3451 was not previously found, it is perhaps because the StaR conformation does not allow binding to the second site, or the second site is only accessible upon proteolytic cleavage of the N terminus. A number of studies have highlighted differences in activation of PAR2-mediated signalling by the tethered ligand compared to peptides[3,37,38]. It is possible that the movement of the tethered ligand into the binding site induces further conformational changes in the receptor, which might have a profound effect on the outside of the helical bundle where the high-affinity binding site of AZ3451 is located, explaining why AZ3451 induces partial inhibition of only the tethered ligand-activated state. A recent paper reported a similar finding, a PAR2 antagonist attached to a PEG linker partially inhibited SLIGKV-induced activation of Ca$^{2+}$ signalling via PAR2, with IC$_{50}$ = 0.1 μM, however, had no effect on trypsin-induced activation of PAR2[39]. Antagonists that can distinguish between peptide and protease-induced activation of PAR2 provide opportunities to improve our understanding of the endogenous mechanism of activation by proteases.

In summary, we describe the discovery of novel, potent, selective agonists and antagonists for PAR2, which allowed the identification of PAR2 signalling from discrete ligand binding

sites. Binding studies and pharmacological characterisation have determined two distinct mechanisms of inhibition, orthosteric for an imidazole AZ8838 and allosteric for a benzimidazole AZ3451. Both compounds showed activity at human and rat PAR2 in a range of different cell types. We used an acute inflammation model[30], oedema reported to be induced in rat paws by PAR2 peptide agonists such as 2f-LIGRLO-$NH_2$, GB110, SLIGRL-$NH_2$ and the endogenous agonist trypsin, to investigate anti-inflammatory activity for the new PAR2 antagonists. Pre-treatment with either AZ8838 or AZ3451 significantly reduced agonist-induced paw swelling and inhibited mast cell and neutrophil activation and degranulation in this model. Thus, both PAR2 inhibitory mechanisms have disease-modifying anti-inflammatory properties in vivo that can be used as a basis for further design and development of PAR2 targeted therapeutics.

## Methods

**Materials**. SLIGKV-$NH_2$ and SLIGRL-$NH_2$ were custom synthesised by ThermoFisher Scientific. 2f-LIGRLO-$NH_2$ and GB110 were synthesised according to Barry et al.[29] and Eu-tagged-LIGRLO(dtpa)-$NH_2$ was synthesised according to Hoffman et al.[40]. [$^3$H-GB110] was synthesised as in Cheng et al.[24]. Recombinant human trypsin (referred to as trypsin) was from Polymum Scientific. Chemical synthetic routes of compounds and analytical data are described in supplementary methods.

**Europium-tagged competition binding assay**. Competition binding experiments against Eu-tagged-2f-LIGRLO(dtpa)-$NH_2$ were performed as previously described[38]. CHO-hPAR2 cells were seeded overnight in 384-well plate at $2.5 \times 10^4$ cells/well and blocked with 2% BSA for 60 min. Cells were simultaneously exposed to both Eu-tagged-2f-LIGRLO(dpta)-$NH_2$ (300 nM) and test compounds (AZ3451 or AZ8838) for 15 min. Cells were washed 3× with PBS supplemented with 20 μM EDTA, 0.01% Tween and 0.2% BSA followed by addition of DELFIA Enhancement Solution (Perkin Elmer) for 90 min. Time-resolved fluorescence was measured with PHERAstar FS reader (BMG Labtech).

**Radiolabelled competition binding assay**. Competition binding experiments against radiolabel [$^3$H]-GB110 were carried out on membranes isolated from HEKexpi293F cells expressing human PAR2. In 96-well plates, at final assay concentration stated in parenthesis, protein (30 μg per well), label (25 nM), and test compounds AZ3451, AZ8838, AZ2429 (0.6 nM–100 μM), SLIGKV-$NH_2$ (0.3 nM–55 μM) or unlabelled GB110 (0.3 nM–10 μM) were incubated for 2 h at room temperature with shaking. The reaction was then filtered through 0.5% PEI-coated filter plates using a BiomekFX. The filter plate was then dried for 1 h at 50 °C. Scintillation fluid was added to each well, before reading the plate in a Microbeta reader.

**Cell culture**. Cell culture reagents were purchased from ThermoFisher Scientific unless otherwise stated. The parental 1321N1 cell line was obtained from ECACC. 1321N1 cells, stably expressing the human PAR2 receptor, (1321N1-hPAR2) were routinely cultured in Dulbecco's Modified Eagle Medium (DMEM) supplemented with GlutaMax-1 containing 10% fetal bovine serum (FBS) and G418 (600 μg ml$^{-1}$). U2OS cells, stably expressing human PAR2 (U2OS-hPAR2, 93-0235C3, DiscoverX/Eurofins) were routinely cultured in McCoy's 5A media containing 10% FBS, G418 (500 μg ml$^{-1}$) and Hygromycin B (250 μg ml$^{-1}$). Rodent PAR2 was assessed in rat KNRK cells (ATCC) cultured in DMEM with 10% FBS supplemented with 2 mM L-Glutamine. Human PAR2 was assessed in HT-29 cells (ECACC) cultured in DMEM with 10% FBS supplemented with penicillin (50 U ml$^{-1}$) and streptomycin (50 μg ml$^{-1}$). Flp-In CHO-hPAR2 cells were generated as described[27], pcDNA5/FRT vector containing hPAR2 was cotransfected with pOG44 Flp-recombinase expression vector using Lipofectamine 2000 and maintained in Ham's F-12 media supplemented with 10% FBS, 2 mM L-Glutamine and 200 μg ml$^{-1}$ Hygromycin B. All cells were tested to be mycoplasma free and verified by STR fingerprinting. For assays, cells were seeded in the respective media without selection antibiotics. 1321N1 cells transiently-transfected with wild type (WT) or single-point mutants of PAR2 (1321N1-PAR2[WT]; 1321N1-PAR2[F154A]; 1321N1-PAR2[G157C]), were generated as described[24], thawed and plated directly for assays.

**$Ca^{2+}$ flux assay**. For evaluation of $Ca^{2+}$ mobilisation, 1321N1-hPAR2 cells were seeded in 384-well CellBIND plates (Corning) at 4000 cells per well in DMEM containing 10% FBS and incubated for 18–24 hours at 37 °C, 5% $CO_2$ and 95% humidity. Cells were loaded with Fluo-8 NW calcium dye (AAT Bioquest) for 30 min at 37 °C before compounds in Hank's balanced salt solution (HBSS, ThermoFisher Scientific) with 20 mM HEPES (ThermoFisher Scientific), 0.1% bovine serum albumin (BSA, Sigma) and 0.5% DMSO were added and simultaneously evaluated for changes in intracellular [$Ca^{2+}$] using FLIPR TETRA

(Molecular Devices) to detect agonism. For each concentration, the difference between the maximum peak $Ca^{2+}$ response and baseline response is reported. Evaluation of antagonism was performed after an additional 30 min incubation at room temperature followed by addition of SLIGRL-$NH_2$ (2 μM), trypsin (39 nM) or tryptase (1 μM) at $EC_{80}$ for activation of PAR2 and detection of $Ca^{2+}$ flux. Antagonism experiments were also carried out on 1321N1-PAR2[WT], 1321N1-PAR2[F154A], and 1321N1-PAR2[G157C] as described. Experiments with trypsin in 1321N1-hPAR2 cells were performed in the presence of 1 μM Vorapaxar (Chemtronica) to remove any interference of PAR1 activity. Inhibition of $Ca^{2+}$ release on rat PAR2 was measured in the same way using 10 000 KNRK cells per well seeded in poly-D-lysine coated 384-well plates (Greiner), 30 min preincubation with the compound before activation of PAR2 with $EC_{80}$ of SLIGRL-$NH_2$ (120 μM) and real-time detection of $Ca^{2+}$. Antagonism experiments for CHO-hPAR2 and HT-29 cells were performed in 96-well plates at 30,000 cells/well. Cells were loaded with Fluo-3 AM calcium dye (Sigma-Aldrich) for 60 min at 37 °C in HBSS with 1% FBS, 0.2% pluronic acid, 20 mM HEPES and 2.5 mM probenecid prior to the addition of antagonist. The antagonist was pre-incubated for 60 min at 37 °C followed by the addition of bovine trypsin (30 nM) at $EC_{80}$. Intracellular [$Ca^{2+}$] flux was measured using FLIPR TETRA (Molecular Devices) and the difference between the maximum peak $Ca^{2+}$ response and baseline response is reported.

**IP1 assay**. Inhibition of PAR2-induced inositol-1-phosphate (IP1) production was measured in 1321N1-hPAR2 cells using IP-One $G_q$ HTRF assay kit (Cisbio). Compounds were pre-incubated with 15 000 cells per well in white small-volume 384-well plates (Greiner) for 30 min at 37 °C in 20 mM HEPES with HBSS. Next, $EC_{80}$ of SLIGRL-$NH_2$ (70 μM) supplemented with 50 mM LiCl was added and incubated for 60 min at 37 °C. Detection of IP1 production was performed according to the manufacturer's protocol and HTRF was measured using PHERAstar (BMG LabTech). Data were normalised to [IP1] using a standard curve. Activation of PAR2 and subsequent IP1 production was measured in a similar way, but with 60 min compound incubation in the presence of 50 mM LiCl before the detection of IP1.

**β-arrestin-2 assay**. β-Arrestin-2 recruitment was measured in U2OS-hPAR2 cells using the DiscoverX assay kit, according to the manufacturer's instructions. Briefly, cells were seeded in McCoy's 5A media containing 10% FBS at 4000 cells per well in 384-well plates and incubated overnight at 37 °C, 5% $CO_2$. Antagonist compounds (30 μM–1 nM) were incubated with the cells for 30 min prior to the addition of agonist SLIGRL-$NH_2$ (10 μM) at $EC_{80}$ and further incubation for 90 min at 37 °C. Working detection solution (12 μl) was then added to each well, incubated at room temperature in the dark for 60 min and the chemiluminescence signal read on the EnVision microplate reader (Perkin Elmer).

**pERK1/2 assay**. Phosphorylation of ERK1/2 was measured in U2OS-hPAR2 cells using Phospho-ERK1/2 (Thr202/Tyr204) assay kit (64ERKPEH Cisbio) and the 2 plate manufacturer's protocol with the following modifications. Cells were seeded in McCoy's 5A media containing 10% FBS at 15,000 cells per well in 384-well plates and incubated overnight at 37 °C, 5% $CO_2$. Antagonist compounds (30 μM–1 nM) were incubated with the cells for 30 min prior to the addition of agonist SLIGRL-$NH_2$ (10 μM) at $EC_{80}$ and further incubation for 5 min at 37 °C. The supernatant was removed from the wells and lysis buffer was added to the cells and incubated with shaking at room temperature for 30 min. The cell lysate was transferred into a small-volume 384-well plate containing detection antibodies and incubated for 4 h at room temperature in the dark. Plates were read for HTRF signal on PHERAstar (BMG LabTech). In addition, phosphorylation of ERK1/2 in CHO-hPAR2 using AlphaLISA SureFire Ultra pERK1/2 (Thr202/Tyr204) (Perkin Elmer). Cells were seeded at a density of 2000 cells per well in a 384-well proxiplate and incubated overnight. Antagonists were pre-incubated for 30 min at 37 °C prior to the addition of agonist (1 μM 2f-LIGRL-$NH_2$ or 50 nM trypsin) for 10 min at 37 °C. After incubation, cell lysis buffer was added with shaking for 30 min. Antibodies and beads were added and incubated for 4 h at room temperature. AlphaLISA signal was measured using a PHERAstar (BMG LabTech).

**Data analysis**. Binding data were analysed using the One site – Fit $K_i$ model in GraphPad Prism 8 (GraphPad Software) to determine binding affinity. Pharmacology in vitro potency data was analysed with a 4-parameter logistic fit using the equation $y = \text{Bottom} + (\text{Top} - \text{Bottom})/(1 + 10)\wedge((\text{LogXC50-x})*(\text{HillSlope}))$ where y is the response, x is the 10-logarithm of the drug concentration, Bottom is no response and Top is full response in GraphPad Prism 8 or Genedata Screener (Genedata). $pEC_{50}$ and $pIC_{50}$ data are presented as mean ± standard error of the mean (s.e.m.) derived from the number of independent experiments stated. Schild slope analysis was calculated using Graphpad Prism 8 Gaddum/Schild $EC_{50}$ global fitting (shared model parameters between all data sets) competitive model. The most rigorous method for quantifying agonist-antagonist interactions is to globally fit the Gaddum/Shild model to all agonist dose–response curves[41]. Schild slope quantifies the shifts in agonist response in the presence of antagonist, a Schild slope of 1 indicates the antagonist is competitive and extra-sum-of-squares $F$-test ($p <$ 0.05) was used to compare the calculated Schild slope against a hypothetical Schild slope of 1. Allosteric interaction studies between ligands in the calcium flux assay

were fitted to the following operational model of allosterism[42]:

$$E = \frac{E_m[\tau_A[A](K_B + \alpha\beta[B]) + \tau_B[B]K_A)]^n}{([A]K_B + K_A K_B + K_A[B] + \alpha[A][B])^n + [\tau_A[A](K_B + \alpha\beta[B]) + \tau_B[B]K_A)]^n},$$

$E_m$ is the maximum possible cellular response and held constant at 100, [A] and [B] are the concentrations of the orthosteric and allosteric ligands, respectively, $n$ represents the slope of the transducer function and was held constant at 1. $K_A$ and $K_B$ are equilibrium dissociation constant of the orthosteric and allosteric ligands respectively as determined with Eu-tagged-2f-LIGRLO(dtpa)-NH$_2$, p$K_A$ of 2f-LIGRLO-NH$_2$ = 6.1, p$K_B$ of AZ3451 = 6.9. Parameters $\tau_A$ and $\tau_B$ denote the capacity of orthosteric and allosteric ligands, respectively, to exhibit agonism, log$\tau_A$ = 0.7 and log$\tau_B$ = 0 since AZ3451 showed no agonist activity. $\beta$ represents the efficacy factor between orthosteric and allosteric ligands. $\alpha$ represents the cooperativity factor between orthosteric and allosteric ligands (values of $\alpha > 1$ denote positive cooperativity, $\alpha < 1$ denote negative cooperativity and $\alpha = 0$ denote neutral cooperativity). Allosterism for AZ3451 in the calcium flux assay was determined and analysed. The relative activity was calculated using $(E_{max}EC_{50})/(E_{max}'EC_{50}')$ with the Hill slope help constant at 1. $EC_{50}'$ and $E_{max}'$ denote the $EC_{50}$ and $E_{max}$ in the absence of allosteric modulator, while $EC_{50}$ and $E_{max}$ denote those measured in the presence of allosteric modulator. When the Hill slope = 1, the relative activity plot approaches an asymptote corresponding to the estimate of the $\gamma$ value.

**Molecular modelling.** AZ2429 was prepared using Ligprep[43]. Docking of AZ2429 was performed using Glide SP[44] to a refined PAR2 model previously described[26]. In short, the PAR2 complex structure 5NDD[24] was refined to fit peptide ligands and peptidomimetics by placing GB88 in the proposed binding site and creating a new model using the 5NDD structure as a template in Prime[45] to allow for side-chain rearrangements. Ligand placement during docking was guided using a positional constraint of a phenyl moiety in place of the phenyl ring of AZ8838 from the PAR2 complex structure. Positional constraints were used with a tolerance of 2.4 Å.

**Surface plasmon resonance (SPR) binding.** The SPR binding experiments were performed on a Biacore S200 optical biosensor unit (GE Healthcare) at 30 °C. Sensor chips Series S NTA (Research grade, GE Healthcare) were equilibrated at room temperature prior to use. The running buffer for protein tethering and subsequent ligand binding experiments was 50 mM HEPES, 250 mM NaCl, 0.1% (w/v) LMNG, 0.005% (w/v) CHS, 0.025% (w/v) CHAPS, pH 7.4, if not stated otherwise. The tethering of PAR2 was performed at a flow rate of 5 μl min$^{-1}$. The NTA surface was chosen to allow full control of the steric orientation of the tethered protein and was conditioned by a one-minute injection of a 350 mM EDTA solution (pH 8.3) followed by a one-minute injection of running buffer supplemented with 0.5 mM NiCl$_2$. To allow for a covalent tethering after the initial capture step, the conditioned surface was activated for 7 min with 50 mM NHS and 200 mM EDC. This was immediately followed by an injection of PAR2 in running buffer at a concentration of 80–120 μg ml$^{-1}$ with a contact time of 5–10 min to achieve the desired densities of 8000–9000 RU. The deactivation of residual esters was achieved through a waiting period of 4–5 h that also resulted in a more stable SPR signal. Reference surfaces were prepared accordingly, omitting the injection of protein over the activated reference surface. The subsequent binding experiments were all performed at a flow rate of 30 μl min$^{-1}$ and by employing the method of single-cycle kinetics. This approach involves the sequential injection of a compound concentration series without regeneration steps. A contact time of 60 s was selected, which was followed by a 30 min dissociation phase to allow for a proper estimation of the dissociation rate constant. Compounds were dissolved in DMSO to 10 mM and a digital dispenser HP D300 (Tecan) was used to set up the compound concentration series using 8 concentrations with a 2-fold dilution pattern. The tested concentrations for AZ3451 and AZ2429 have been 8, 16, 32, 64, 128, 256, 512, 1024 nM and 75, 150, 300, 600, 1200, 2400, 4800, 9600 nM, respectively. Prior to the analysis of compound binding, three running buffer blanks were injected to equilibrate the instrument. The data collection rate was set to 10 Hz and all experiments were repeated at least four times to allow for error estimations. Due to the low DMSO mismatch (maximum 0.1%) introduced by the compound addition, no solvent correction was required. To evaluate compound binding to the PAR-AZ8838 complex, the antagonist AZ8838 was supplemented into the running buffer at a concentration of 50 μM and the instrument was equilibrated for 1 h before performing the single-cycle kinetic study as described above. Similarly, binding of AZ2429 to the PAR-AZ3451 complex was investigated after addition of 1 μM AZ3451 to the running buffer followed by 1 h equilibrium period, while the addition of 10 μM AZ2429 to the running buffer was required to ensure >99% saturation for single-cycle kinetic study of AZ3451 binding to the PAR2-AZ2429 complex.

**Data fitting and statistical analysis of SPR binding experiments.** Prior to the fitting of the data, the reference-subtracted data was further analysed by subtracting a similar single-cycle kinetic experiment containing only buffer injections in order to correct for injection artefacts, systematic noise and instrument drift. This double-referenced data has been fitted using a simple 1:1 interaction model as described by Karlsson et al.[46] in order to extract kinetic and affinity data. All data

were expressed as mean ± s.e.m. and further analysed using GraphPad Prism 8. Statistical differences were assessed using student's t-tests (two-tailed) for data sets of two and one-way ANOVA for data sets of three. Significance was set at $^+p < 0.05$, $^{++}p < 0.01$, $^{+++}p < 0.001$, $^{++++}p < 0.0001$ compared to AZ3451, and $^*p < 0.05$, $^{**}p < 0.01$, $^{***}p < 0.001$, $^{****}p < 0.0001$ compared to AZ2429.

**In vivo studies on PAR2 antagonists.** Male Wister rats (7–8 weeks old, 250 ± 30 g) were bred at the Animal Resources Centre (Canning Vale, Australia). Following Australian ethical standards, animals were air transported to the Institute for Bioengineering and Nanotechnology (University of Queensland, Australia). Animals were housed at room temperature and kept in 12 h light/dark cycles, with standard chow and water provided ad libitum, for at least a 48 h acclimatisation period before experiments. Hind paw swelling was induced by i.pl. injection of 2f-LIGRLO-NH$_2$ at 350 μg/100 μl saline/paw. Saline only was given to sham paws. Treated animals received AZ3451 (in DMSO, 10 mg ml$^{-1}$ kg$^{-1}$ s.c.) or AZ8838 (in 20% DMSO/80% olive oil, 10 mg ml$^{-1}$ kg$^{-1}$ p.o.) 30 min or 2 h prior to administering 2f-LIGRLO-NH$_2$, respectively. Control animals received either DMSO s.c. or 20% DMSO/80% olive oil via oral gavage. All injected paws were measured (width, thickness) with a digital calliper (World Precision Instruments, Sarasota, FL, USA) just before 2f-LIGRLO-NH$_2$ or saline administration (0 h), as well as at 0.5, 1, 2 and 4 h post-oedema induction. Data were expressed as a normalised % change in cross-sectional area (mm$^2$) from the baseline against maximal swelling at 30 min induced by 2f-LIGRLO-NH$_2$ alone. At the 30 min time point, a portion of animals was euthanised by CO$_2$ inhalation and their paws collected for further analyses. All in vivo experimental procedures and animal handling were conducted with approval from the Animal Ethics Committee at the University of Queensland which follows NHMRC and ARRIVE guidelines. After experimentation, animals are humanely euthanized by CO$_2$ inhalation as stipulated by the ethical agreements.

**Histopathology and immunohistochemistry.** Rat paw tissues were collected, fixed in 10% neutral-buffered formalin (Sigma-Aldrich) for 2 h at 4 °C, transferred into falcon tubes containing 25% sucrose in PBS and left overnight at 4 °C. Tissues were then embedded with Optimal Cutting Temperature compound (OCT, Sakura Finetek, USA) and stored at −80 °C before making cryo-sections for histological analysis. Frozen sections (5 μm) were made using a cryostat-microtome (Leica Biosystems, Germany). For mast cell staining, slides were briefly washed with distilled water before staining with 0.1% toluidine blue (Sigma-Aldrich, pH 2.5) for 3 min. Slides were then dehydrated and mounted in DPX mountant (Sigma-Aldrich, USA). For immunohistochemistry, slides were briefly rinsed with PBS to remove OCT. Except for tryptase staining, antigen retrieval was performed by incubating slides in citrate buffer (0.01 M, pH 6) at 95 °C for 20 min. They were then cooled down briefly and incubated with blocking medium (PBS, 0.1% Triton X-100, 10% horse serum) for 1 h at room temperature in a humidity chamber. Samples were incubated with primary antibody medium (PBS, 0.1% Triton X-100, 4% horse serum; 1:100 primary antibody including – histamine, neutrophil elastase or ED1 antibodies) overnight at 4 °C. The next day, samples were washed with PBS and incubated with the secondary antibody medium (PBS, 0.1% Triton X-100, 4% horse serum, detection antibody 1:200) for 2 h at room temperature. For tryptase staining, slides were incubated with primary antibody medium (as above with tryptase antibody at 1:100 dilution) for 4 h, then the secondary antibody medium for 2 h. Finally, slides were dried, counterstained with DAPI (Invitrogen, Australia), and sealed with clear nail polish. Primary antibodies for tryptase (AA1, ab2378, Abcam, Australia), histamine (H7403, Sigma-Aldrich, USA), neutrophil elastase (ab21595, Abcam, Australia) and ED1 (ab31630, Abcam, Australia) were purchased from commercial sources. Secondary detection antibody was purchased from Abcam, Australia. All microscopic images were obtained using an Olympus BX-51 upright microscope with Olympus DP-71 12Mp colour camera, utilising DP Capture and DP Manager software packages (Olympus, Tokyo, Japan). Tryptase-, neutrophil elastase- and ED1-positive cells were quantified using FIJI/ImageJ 1.42q software, USA.

**Myeloperoxidase assay.** Rat paw tissues were collected, snap-frozen with liquid N$_2$, and stored at −80 °C. Samples were cut with a scalpel, weighed, and added to MPO buffer (50 mM potassium phosphate buffer, pH 6.0) to a concentration of 200 mg ml$^{-1}$, before homogenising using zirconium oxide beads (1 and 2 mm, Next Advance, USA) and a Bullet Blender® homogeniser (Next Advance, USA). Homogenised tissues were diluted to 100 mg ml$^{-1}$ with MPO buffer containing 1% hexadecyl trimethylammonium bromide (Sigma-Aldrich, USA). Samples were then sonicated for 5 min, centrifuged at $13,000 \times g$ for 10 min at 4 °C, and the clear supernatant for each sample was collected. For MPO release, supernatant (30 μl) was added to 200 μl of MPO buffer containing 0.167 mg ml$^{-1}$ $o$-dianisidine HCl (Sigma-Aldrich USA) and 0.001% H$_2$O$_2$ in a clear 96-well plate, which was read immediately at 460 nm every min for 20 min (PHERAstar FS plate reader, BMG Labtech, Germany).

**Statistical analysis of in vivo and ex vivo studies.** Experimental results were expressed as mean ± s.e.m. Graphs were plotted and data analysed using GraphPad Prism7. Statistical differences were assessed using Student's t-tests (two-tailed) for data sets of two, one-way ANOVA for data sets of three or more with Bonferroni

post-hoc tests, or two-way repeated-measures ANOVA with Tukey post-hoc tests for temporal data sets, as appropriate. Significance was set at $+p < 0.05$, $++p < 0.01$, $+++p < 0.001$, $++++p < 0.0001$ compared to sham, and $*p < 0.05$, $**p < 0.01$, $***p < 0.001$, $****p < 0.0001$ compared to disease control.

**Statistics and reproducibility**. Pharmacological data are presented as mean ± s.e.m. of the indicated number of replicates stated in the relevant figure legend or data table. Graphs show individual data points (from at least two replicates) with error bars denoting s.e.m. Replicates are defined as independent repeats of the experimental protocol. Statistical analyses were conducted as outlined in the relevant method section.

**Reporting summary**. Further information on research design is available in the Nature Research Reporting Summary linked to this article.

## Data availability
The authors declare that the data supporting the findings of this study are available within the article and its Supplementary Information files. The source data for the graphs in the main figures are available as Supplementary Data 1. All the other data supporting the findings of this study are available from the corresponding author upon reasonable request.

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

## Acknowledgements

A.J.K. is a fellow of the AstraZeneca postdoctoral programme. The University of Queensland researchers acknowledge support to D.P.F from NHMRC for a Senior Principal Research Fellowship (1117017) and grants (1047759, 1084083), and from the ARC Centre of Excellence in Advanced Molecular Imaging (CE140100011).

## Author contributions

A.J.K., L.S., Y.J. and J.L. performed pharmacological evaluations in vitro. S.G. performed biophysical characterisation. R.C. and D.G.B. led the chemistry resulting in the discovery of AZ8838. E.K.Y.P and R.-J.L. performed rat paw oedema studies and pharmacological profiling in vivo. A.N. performed molecular modelling of AZ2429. Q.L., W.Y. and R.C. performed the synthesis and chiral purification of AZ2429. M.S. performed the vibrational circular dichroism calculations to establish the absolute stereochemistry of AZ2429. S.J. synthesised AZ0107. A.J.K., L.S., S.G., A.M., D.P.F., D.G.B. and N.D. wrote the manuscript and all authors edited it. D.P.F. led the studies at the University of Queensland; N.D. and D.G.B. were responsible for the overall project strategy.

## Competing interests

AstraZeneca authors are current or former salaried employees of AstraZeneca and have/had stock or stock options in AstraZeneca. The authors declare no other competing interests in these particular studies. The remaining authors declare no competing interests.
