## [Peer Review File · Communications Biology]

Reviewers' comments:

Reviewer #1 (Remarks to the Author):

A) Summary of work

This paper from Dekker and colleagues describes the detailed pharmacological characterization of several previously disclosed PAR2 antagonists and an agonist, using biochemical (SPR) and cell assays examining numerous downstream signals, and also in vivo using rat models of inflammation. The presented data is extensive (including useful SAR studies), the experiments appear to be well designed and performed, and the authors provide detailed evidence for allosteric effects. Interestingly, the authors also observed biphasic antagonism of trypsin-driven PAR2 calcium mobilization by the allosteric ligand AZ3451, which was not observed with other PAR2 agonists. They additionally conclude that both the orthosteric and allosteric PAR2 antagonists are anti-inflammatory in vivo.

B) Impact, quality of research, and overall recommendation

PAR2 is an important therapeutic target for a variety of inflammation-related conditions, and this work is a valuable extension of the innovative structural and med. chem. work from the AstraZeneca/Heptares/X-Chem team (joined here by the Fairlie lab team). I recommend publication of this impressive body of work (which certainly extends far beyond a communication in scope), with the issues below addressed. The measurements of allosterism with the different ligands, and the biphasic responses observed with trypsin is particularly notable and worthy of publication in this venue, though it may receive less notice here than in a more traditional pharmacology/medicinal chemistry journal.

C) Recommended revisions

Major:

1) The protocol for the preparation of AZ2429 is incomplete. Flash chromatography methods are not disclosed, and the methods for HPLC purification are not given (column, solvent mix, etc.).

2) The synthetic route to AZ2429 molecule may also lead to unexpected epimerization of the stereocenter of SM1. Do the authors have evidence that the methyl ester 2 is enantiopure? Did they check for racemization after reaction with thionyl chloride? I would have recommended a milder protection of the acid, such as with Alloc. The hydrolysis of the ester with LiOH should be okay, but could still conceivably epimerize the stereocenter. Additionally, it is odd that the relative configurations of the stereocenters in SM2 are not disclosed. Is this building block actually a mixture of the 4 possible stereoisomers? Most importantly, I am not aware that absolute and relative configurations can be reliably determined using "standard" vibrational circular dichroism methods. Details of the vCD measurements should be given, and in this reviewer's opinion, the absolute configuration of AZ2429 should be clearly described in the manuscript as tentative, until X-ray data can be obtained.

Minor:

3) I'm not sure if "crosstalk" in the title is the right word for the allosterism observed in this paper. I would recommend that "crosstalk" be replaced with "allosterism" in the title.

4) Line 232: How does this allosterism compare to that with previous PAR2 agonists, such as SLIGKV?

5) Line 296: "Remarkably, unlike other PAR2 agonists, AZ2429 can bind to the mutated PAR2 StaR." A little more explanation of StaR, and this result, is justified. Why can't binding of other PAR2 agonists be detected?

6) Line 315: "Thal et al. suggest that upon binding to its site between TM2–4, AZ3451 could prevent the rotation and movement of TM3 required for receptor activation." I think this is the wrong citation– ref 34 is from 2007, and doesn't seem to be about PAR2 structural info.

Reviewer #2 (Remarks to the Author):

The manuscript by Kennedy et al. describes the in vitro and in vivo characterisation of two novel small molecule PAR2 inhibitors and an agonist, which were recently discovered and represent novel tool compounds for studying PAR2. The authors conclude that while the small molecule agonist (AZ2429) and one of the inhibitors (AZ8838) bind to the orthosteric PAR2 binding site, the other inhibitor (AZ3451) is a negative allosteric modulator. While these findings are novel because PAR2 inhibitor discovery has been extremely limited, I have a number of very major concerns with the analysis of data and conclusions drawn from this study, as outlined below.

1. Throughout the results, the authors do not perform statistical analyses on their data. For instance, page 5, the authors state "the benzimidazoles (AZ3451 $K_i = 120$ nM) bound with higher affinity than the imidazoles (AZ8838 $K_i = 3.6$ μ M)"..... "with the benzimidazole series (AZ3451, $IC_{50} = 2.8$ nM) more potent than the imidazole series (AZ8838, $IC_{50} = 2.0$ μ M)", but there are no stats to back up these conclusions. Statistics should be performed on the negative logarithm of the K_i or IC_{50} , because K_i and IC_{50} values are not normally distributed. Similarly, page 8, in reference to the SPR-based binding kinetics of AZ3451 in the presence of AZ2429, the authors state that AZ3451 "rates of association to and dissociation from the receptor were accelerated, and the combination resulted in a net 2-fold reduction in the affinity." However, once again no statistical analysis has been performed to compare the pK_d of AZ3451 in the absence and presence of AZ2429. Given that only an n of 2 has been performed for these experiments, I am not convinced that 14 ± 3 nM (AZ3451 K_d) and 28 ± 5 nM (AZ3451 K_d in the presence of AZ2429) are statistically different, particularly as the authors state that there is no effect of AZ8838 on AZ3451, where AZ3451 K_d in the presence of AZ8838 is 26 ± 1 nM. Further, where an n of 2 has been performed, S.D. should be stated, not S.E.M.

2. SPR is used to measure agonist (AZ2429) and NAM (AZ3451) association and dissociation kinetics (Figures 3b-g). However, the methods for SPR experiments are unclear and it is difficult to determine what concentrations of each ligand have been used in these experiments. I think in Figure 3d, 50 μ M AZ8838 has been added prior to the addition of AZ3451, but there is no information regarding the concentration of AZ3451 used in the experiment shown in Figure 3d. Similarly, I think 5 μ M AZ3451 or AZ2429 was added in Figures 3f and g, respectively, followed by varying concentrations of AZ2429 or AZ3451 (shown in the insert to Figures 3f and g). However, what concentration of these ligands was used in the experiment shown in the main graph for Figures 3f and g (ie. the grey trace?).

3. What equation has been used to analyse the SPR kinetics data? The authors state that a 1:1 binding model was used, but I do not know what this model is referring to and it is not explained in the methods.

4. The authors need to explain what an A-B-A injection scheme is. This sounds like something specific to the Biocore software, but not something that a non-specialist instinctively knows.

5. The authors state that in the presence of AZ8838 "AZ3451 was still able to bind to the receptor

with similar binding kinetics (Figure 3d, Table 3)". However, the data shown in Figure 3b and d look like the time taken to reach AZ3451 equilibrium binding has decreased in the presence of AZ8838. Further, the AZ3451 dissociation kinetics look faster in the presence of AZ8838, which would explain a slowing of equilibrium binding without an effect on K_{on} (Table 3). Is the SPR data shown a representative experiment? Given that these experiments are an n of 2, I am not convinced by this data.

6. With reference to the SPR studies shown in Figure 3e, the authors conclude that "when AZ8838 was bound to the receptor, AZ2429 was no longer able to bind, consistent with overlapping binding sites." However, this experiment does not confirm overlapping binding sites, as a negative allosteric modulator could prevent the binding of a ligand. Only a change in the association or dissociation RATES proves that there is an allosteric interaction.

7. Page 6, the authors state that "AZ2429 was able to fully displace the probe ($K_i = 63$ nM, Figure 2b, Table 2)", however, none of the ligands shown in Figure 2b fully displace [3H]-GB110 binding, with all displacing a maximum of 75%. What is the explanation for this? Further, why is specific binding in this figure greater than 100%?

8. Page 6, the authors describe how mutating residues G157 and F154 (Supplemental Figure 4) disrupts the biphasic response to AZ3451 suggesting "that AZ3451 binds to a high affinity site on PAR2, likely that observed in the crystal structure, while the biphasic behaviour is consistent with a second binding site with lower affinity for this ligand.". However, data points in Figures 4b and c lie outside the regression analysis, and it is therefore not clear whether the biphasic response really is abolished, or whether the error in the data just make it appear so.

9. Page 7 – what do the authors mean by "off-DNA small molecule"? Do they mean synthetic?

10. Page 9 – there is a brief mention that interaction data between AZ3451 and 2f-LIGRL-NH2 or trypsin were fitted to an allosteric model to calculate cooperativity values. However, the model fits to the data should be shown, plus inclusion of a table showing all the parameters in the equation shown on page 17 (i.e. pK_A , pK_B , Logtau_A , Logtau_B , n , α , β , E_m , Basal), including any constraints that were applied during the analysis.

11. For the Schild analysis in Figure 4, it appears that the agonist E_{max} has been shared between curves. What was the rationale for this (ie. was an F test used to determine whether data were fitted best when the E_{max} was shared?). It is clear that at least in panel d, AZ3452 reduces the E_{max} of trypsin.

12. Page 10, discussion. The authors state that AZ3451 "was active in vivo in rats after subcutaneous injection but not after oral delivery." Were the experiments following oral delivery performed in the current study but not shown or previously published?

13. Consistent with Tables 1 and 2, Table 3 should show pK_d , not K_d .

14. In Supplementary Figure 3e, AZ8838 in pERK1/2 assays does not appear to be approaching 100% inhibition, meaning it looks allosteric not competitive. If AZ8838 is truly competitive, it should show similar potency in all inhibition assays providing the same concentration of orthosteric agonist is present. Clearly either its potency or efficacy as an antagonist is reduced in pERK1/2 assays against trypsin. Any explanation?

Reviewer #3 (Remarks to the Author):

In this manuscript the authors expand on recent work from their laboratories to provide a handful

of novel small molecule protease-activated receptor-2 (PAR2) binding compounds. They provide details on 6 antagonists (4 novel) from two distinct "series" and a novel antagonist. The authors do a nice job evaluating multiple in vitro signaling assays including: Ca²⁺ after activation of either human or rat receptor; inositol trisphosphate (IP₁) as a Gq surrogate; pERK1/2 as a mitogen activated p kinase surrogate; and beta-arrestin; that, taken together, helps to clarify full agonist/antagonist potential. The authors additionally use a competitive binding assay, surface plasmon resonance (SPR) and single cycle kinetics followed by pharmacological analyses to evaluate compound interaction with PAR2. The authors finish their report with an in vivo response using a straight-forward paw oedema model for inflammation associated with disease. The authors argue that differences in pharmacology allow for a novel interpretation of at least two distinct binding sites for antagonists can allow for better bench tools in the laboratory and focus on the development of novel compounds for PAR2 antagonism. In all, the four new antagonists derived from previously published AZ8935 and AZ8838 as well as the novel agonist (AZ2429) are well characterized. There is some concern/confusion, however, as to the novelty of the report in that the two previously published compounds were already characterized as having different binding pockets (e.g., Cheng et al) and it has been proposed that these have different binding sites (what the authors are apparently referring to as "functional cross-talk"). In all, the new antagonist compounds represent important and relatively well-characterized additions for potential use in PAR2 drug discovery.

Major Concerns:

1) Abstract and throughout: it is unclear what is meant by "crosstalk" As far as can be determined from the data presented, the authors are describing two different antagonistic mechanisms/binding sites that have been previously proposed by this group (an orthosteric site and an allosteric site) with two of the compounds used in this manuscript (AZ8838 and A8935; Cheng et al., 2017). In the biological literature crosstalk generally refers to the interaction of two distinct signaling pathways to cause a separate event. While the potential for signal transduction crosstalk is apparent with the functional bias that has been shown to occur with PAR2, the focus of the apparent cross talk described in this manuscript appears to be that "AZ8838 and AZ3451 have distinct modes of action" (Page 8, header). Unless this reviewer misinterpreted the data, the authors are discouraged in using "crosstalk" in describing the differences of orthosteric and allosteric inhibition. A proper "crosstalk" evaluation would include comparison of single compounds and combinatorial compounds that lead to a different signaling/physiological outcome. It would seem that a different description of the varied binding pockets might be in order to best describe the results presented.

2) AZ3451 and AZ8838 have already been reported in at least two separate articles from this group (Cheng, et al., 2017; Brown et al., 2018). These articles already describe the different proposed orthosteric and allosteric binding sites on PAR2. While the authors do clearly cite that these have been published, and there are new compounds presented, in the end it is the originally and previously published compounds that are used throughout as the "novel" compounds (e.g., all in vivo work is done with AZ3451 and AZ8838). Further, it is a bit of a stretch to call a group of two compounds (the "imidazoles" AZ8838 and AZ0107) a "series." In summary there is concern about an overstatement of the novelty of the compounds presented.

3) Binding was mostly assayed with a competitive binding assay but is listed as "Specific Binding" in Table 1 and throughout the manuscript (e.g., page 5, line 110). This should be corrected throughout the manuscript.

4) Activation of PAR2 varies throughout the manuscript. Early in the manuscript comparisons are made to the low potency SLIGVK-NH₂ peptide. However, as the manuscript moves on, other agonists are used, including GB110, 2f-LIGRL-NH₂, 2f-LIGRLO-NH₂. When trypsin is used, a concentration is given rather than the Units of activity. It should be made clear why changes were made or why changes do not matter and how much trypsin is used for activation/antagonism

experiments. This is especially important for readers that have not worked directly with PAR2, but also important for understanding potency/efficacy of antagonists across natural activators (i.e., trypsin units vs other serine proteases).

Minor Concerns:

5) Page 3, line 47: "Protease-activated receptors (PAR)..." should show the abbreviation as "PARs" since throughout PAR refers to the singular.

6) Page 3, line 65: "PAR receptor" is redundant (i.e., protease-activated receptor receptor).

7) Page 5, line 121: "IP-One" should be IP-1, as used throughout.

8) Starting on Page 5, line 123 and throughout manuscript: nM values listed for EC50 do not match with values in Tables 1, 2. The use of two different numerations is confusing (pIC50 vs IC50), but somewhat understandable for ease of use in the Tables. However, conversions do not match. At the least, significant figures should allow for simple conversions. It is also unclear why the authors sometimes use 0.X microM vs x00 nM (e.g., page 7, lines 178 and 180).

9) The authors seem to gloss over the different delivery of AZ8838 and AZ3451 in the evaluation of paw oedema (e.g., page 13, line 372; page 19, lines 545-546). Does this mean that AZ8838 does not work s.c? or AZ3451 not work p.o? The differences should be clearly stated.

Reviewers' comments:

Reviewer #1 (Remarks to the Author):

A) Summary of work

This paper from Dekker and colleagues describes the detailed pharmacological characterization of several previously disclosed PAR2 antagonists and an agonist, using biochemical (SPR) and cell assays examining numerous downstream signals, and also in vivo using rat models of inflammation. The presented data is extensive (including useful SAR studies), the experiments appear to be well designed and performed, and the authors provide detailed evidence for allosteric effects. Interestingly, the authors also observed biphasic antagonism of trypsin-driven PAR2 calcium mobilization by the allosteric ligand AZ3451, which was not observed with other PAR2 agonists. They additionally conclude that both the orthosteric and allosteric PAR2 antagonists are anti-inflammatory in vivo.

B) Impact, quality of research, and overall recommendation

PAR2 is an important therapeutic target for a variety of inflammation-related conditions, and this work is a valuable extension of the innovative structural and med. chem. work from the AstraZeneca/Heptares/X-Chem team (joined here by the Fairlie lab team). I recommend publication of this impressive body of work (which certainly extends far beyond a communication in scope), with the issues below addressed. The measurements of allosterism with the different ligands, and the biphasic responses observed with trypsin is particularly notable and worthy of publication in this venue, though it may receive less notice here than in a more traditional pharmacology/medicinal chemistry journal.

C) Recommended revisions

Major:

1) The protocol for the preparation of AZ2429 is incomplete. Flash chromatography methods are not disclosed, and the methods for HPLC purification are not given (column, solvent mix, etc.).

2) The synthetic route to AZ2429 molecule may also lead to unexpected epimerization of the stereocenter of SM1. Do the authors have evidence that the methyl ester 2 is enantiopure? Did they check for racemization after reaction with thionyl chloride? I would have recommended a milder protection of the acid, such as with Alloc. The hydrolysis of the ester with LiOH should be okay, but could still conceivably epimerize the stereocenter. Additionally, it is odd that the relative configurations of the stereocenters in SM2 are not disclosed. Is this building block actually a mixture of the 4 possible stereoisomers? Most importantly, I am not aware that absolute and relative configurations can be reliably determined using "standard" vibrational circular dichroism methods. Details of the vCD measurements should be given, and in this reviewer's opinion, the absolute configuration of AZ2429 should be clearly described in the manuscript as tentative, until X-ray data can be obtained.

In response to points 1 and 2, we have revised the experimental section for the preparation of AZ2429 to give a detailed procedure from commercially available reagents. The initial sample was identified by

separating the mixture as shown in the scheme. However, new samples were prepared by resolution of an intermediate followed by elaboration to the final compound. Materials from the two routes were compared by chiral chromatography and their biological activities.

We feel that the revised description for the preparation of AZ2429 indicates a clear method to prepare the molecule studied in this paper. The conditions for the separation of the enantiomers are reported and the required enantiomer is identified without the need to know the absolute stereochemistry.

With respect to possible epimerisation during the amide coupling, LC data for the final compound indicated high purity (>99%) when the material was analysed with achiral chromatography. However, chiral chromatography showed the presence of 4.6% of another substance which is consistent with a small amount of epimerisation.

Based on our experience, we feel that VCD analysis can assign relative and absolute stereochemistry with good confidence, but we accept that this cannot be complete. Consequently, we are happy to accept that the stereochemistry is noted as 'tentative' in the absence of an X-ray crystal structure.

For information, we include these background references on the technique:

Applications of Vibrational Spectroscopy in Pharmaceutical Research and Development

- Don E. Pivonka (Editor), John M. Chalmers (Editor), Peter R. Griffiths (Editor)
- May 2007
- ISBN: 978-0-470-87087-7

Vibrational Optical Activity: Principles and Applications

- Laurence A. Nafie
- September 2011
- ISBN: 978-0-470-03248-0

VCD Spectroscopy for Organic Chemists

- Philip J. Stephens, Frank J. Devlin, James R. Cheeseman
- September 2019
- ISBN 9780367381202

Minor:

3) I'm not sure if "crosstalk" in the title is the right word for the allosterism observed in this paper. I would recommend that "crosstalk" be replaced with "allosterism" in the title.

We thank the reviewer for their suggestion and, having considered Reviewer 3 comment 1 as well, we have replaced the title with 'Protease-activated receptor-2 ligands reveal orthosteric and allosteric mechanisms of receptor inhibition'.

4) Line 232: How does this allosterism compare to that with previous PAR2 agonists, such as SLIGKV?

We are unable to measure the coupling of AZ3451 and SLIGKV-NH₂ with SPR, as SLIGKV-NH₂ and other peptide agonists are unable to bind to the mutant thermo-stabilised PAR2 receptor, necessary for this technique. Therefore, we measured allosterism pharmacologically using cells expressing PAR2, and the peptide agonist 2f-LIGRL-NH₂ as well as the protease agonist, trypsin, which cleaves the N-terminus of the receptor to reveal the tethered ligand SLIGKV-. Figure 4 c (peptide) and d (protease) show that AZ3451 is a non-competitive negative allosteric modulator. This is discussed in the final paragraph of section 'Antagonists AZ8838 and AZ3451 have distinct modes of action'.

5) Line 296: "Remarkably, unlike other PAR2 agonists, AZ2429 can bind to the mutated PAR2 StaR." A little more explanation of StaR, and this result, is justified. Why can't binding of other PAR2 agonists be detected?

Further explanation has been added to the manuscript.

'Remarkably, unlike other PAR2 agonists, AZ2429 can bind to the PAR2 stabilised receptor (StaR), which contains a number of mutations to improve its thermostability. The mutations lock the receptor in an antagonist bound configuration, closing off the agonist binding site such that the peptide/tethered ligand agonists are unable to bind.²⁶ Interestingly, agonist AZ2429 is able to bind, thought to be because of its small molecule nature and due to its heterocycle binding in a stable T-shape orientation between H227, D228 and Y82.'

6) Line 315: "Thal et al. suggest that upon binding to its site between TM2–4, AZ3451 could prevent the rotation and movement of TM3 required for receptor activation." I think this is the wrong citation– ref 34 is from 2007, and doesn't seem to be about PAR2 structural info.

Thank you for noticing this, the reference has been changed in the manuscript.

Reviewer #2 (Remarks to the Author):

The manuscript by Kennedy et al. describes the in vitro and in vivo characterisation of two novel small molecule PAR2 inhibitors and an agonist, which were recently discovered and represent novel tool compounds for studying PAR2. The authors conclude that while the small molecule agonist (AZ2429) and one of the inhibitors (AZ8838) bind to the orthosteric PAR2 binding site, the other inhibitor (AZ3451) is a negative allosteric modulator. While these findings are novel because PAR2 inhibitor discovery has been extremely limited, I have a number of very major concerns with the analysis of data and conclusions drawn from this study, as outlined below.

1. Throughout the results, the authors do not perform statistical analyses on their data. For instance, page 5, the authors state "the benzimidazoles (AZ3451 Ki = 120 nM) bound with higher affinity than the imidazoles (AZ8838 Ki = 3.6 μM)" "with the benzimidazole series (AZ3451, IC50 = 2.8 nM) more potent than the imidazole series (AZ8838, IC50 = 2.0 μM)", but there are no stats to back up these

conclusions. Statistics should be performed on the negative logarithm of the K_i or IC_{50} , because K_i and IC_{50} values are not normally distributed.

Statistical analysis of all pharmacological measurements has been performed (on the negative logarithms) and was included in Tables 1 and 2. The authors included only EC_{50}/K_i values in the text to simplify reading for non-pharmacologists. However, as requested by both reviewers 2 and 3, the values within the text have been changed such that both table and text now have $pIC_{50}/pEC_{50}/pK_i$ with errors for consistency.

Similarly, page 8, in reference to the SPR-based binding kinetics of AZ3451 in the presence of AZ2429, the authors state that AZ3451 “rates of association to and dissociation from the receptor were accelerated, and the combination resulted in a net 2-fold reduction in the affinity.” However, once again no statistical analysis has been performed to compare the pK_d of AZ3451 in the absence and presence of AZ2429. Given that only an n of 2 has been performed for these experiments, I am not convinced that 14 ± 3 nM (AZ3451 K_d) and 28 ± 5 nM (AZ3451 K_d in the presence of AZ2429) are statistically different, particularly as the authors state that there is no effect of AZ8838 on AZ3451, where AZ3451 K_d in the presence of AZ8838 is 26 ± 1 nM. Further, where an n of 2 has been performed, S.D. should be stated, not S.E.M.

Thank you for your comments. We have taken this on board and performed further SPR experiments to increase our n numbers and confirm our original findings. Please see the updated Figure 3 and Table 3, which now includes data generated from at least $n=4$ independent replicates, and full statistical analysis of the data.

2. SPR is used to measure agonist (AZ2429) and NAM (AZ3451) association and dissociation kinetics (Figures 3b-g). However, the methods for SPR experiments are unclear and it is difficult to determine what concentrations of each ligand have been used in these experiments. I think in Figure 3d, 50 μ M AZ8838 has been added prior to the addition of AZ3451, but there is no information regarding the concentration of AZ3451 used in the experiment shown in Figure 3d. Similarly, I think 5 μ M AZ3451 or AZ2429 was added in Figures 3f and g, respectively, followed by varying concentrations of AZ2429 or AZ3451 (shown in the insert to Figures 3f and g). However, what concentration of these ligands was used in the experiment shown in the main graph for Figures 3f and g (ie. the grey trace?).

This has been clarified now in the methods section on pages 17-18.

3. What equation has been used to analyse the SPR kinetics data? The authors state that a 1:1 binding model was used, but I do not know what this model is referring to and it is not explained in the methods.

Clarification of the 1:1 interaction model has been added to the methods section (page 19, line 551)

4. The authors need to explain what an A-B-A injection scheme is. This sounds like something specific to the Biocore software, but not something that a non-specialist instinctively knows.

Experiments have been repeated using single cycle kinetics and by providing the compounds in the running buffer. As such the A-B-A injection scheme was no longer required to obtain the data presented in this manuscript and has thus been omitted from the revised version. The methods section now states that all experiments were performed using single cycle kinetics (page 18, line 528–530).

5. The authors state that in the presence of AZ8838 “AZ3451 was still able to bind to the receptor with similar binding kinetics (Figure 3d, Table 3)”. However, the data shown in Figure 3b and d look like the time taken to reach AZ3451 equilibrium binding has decreased in the presence of AZ8838. Further, the AZ3451 dissociation kinetics look faster in the presence of AZ8838, which would explain a slowing of equilibrium binding without an effect on K_{on} (Table 3). Is the SPR data shown a representative experiment? Given that these experiments are an n of 2, I am not convinced by this data.

As stated above, these SPR experiments have been independently repeated at least 4 times to confirm our findings. The new data exemplifying the SPR sensorgrams are representative for all experiments. The additional data is included within the manuscript and the SPR section has been rewritten to better clarify our findings (see Figure 3, table 3 and results section ‘Antagonists AZ8838 and AZ3451 have distinct modes of action’ page 8–9).

6. With reference to the SPR studies shown in Figure 3e, the authors conclude that “when AZ8838 was bound to the receptor, AZ2429 was no longer able to bind, consistent with overlapping binding sites.” However, this experiment does not confirm overlapping binding sites, as a negative allosteric modulator could prevent the binding of a ligand. Only a change in the association or dissociation RATES proves that there is an allosteric interaction.

As stated above, these SPR experiments have been repeated to confirm our findings. The additional data is included within the manuscript and the SPR section has been rewritten to better clarify our findings (see Figure 3, table 3 and results section ‘Antagonists AZ8838 and AZ3451 have distinct modes of action’ page 8–9). As stated correctly by the reviewer, only a change in the binding kinetics would reveal an allosteric coupling. This is in fact observed for the antagonist AZ3451 and the agonist AZ2429 which suggests an allosteric coupling between those binding sites. The complete and efficient prevention of AZ2429 binding to the PAR2-AZ8838 complex is rather a manifestation of a physical interference that we can only explain by overlapping binding sites as shown in Figure 3a.

7. Page 6, the authors state that “AZ2429 was able to fully displace the probe ($K_i = 63$ nM, Figure 2b, Table 2)”, however, none of the ligands shown in Figure 2b fully displace [3H]-GB110 binding, with all displacing a maximum of 75%. What is the explanation for this? Further, why is specific binding in this figure greater than 100%?

We thank the reviewer for spotting this, the data has been re-analysed correctly such that specific binding is measured as a percentage of the total binding minus the non-specific control.

8. Page 6, the authors describe how mutating residues G157 and F154 (Supplemental Figure 4) disrupts the biphasic response to AZ3451 suggesting “that AZ3451 binds to a high affinity site on PAR2, likely that observed in the crystal structure, while the biphasic behaviour is consistent with a second binding site

with lower affinity for this ligand.”. However, data points in Figures 4b and c lie outside the regression analysis, and it is therefore not clear whether the biphasic response really is abolished, or whether the error in the data just make it appear so.

The updated supplementary Figure 4 now shows data over a wider range of concentrations (sub nanomolar to 100uM) with data points describing the full curve with clear biphasic character for wild type PAR2, and monophasic response for point mutations.

9. Page 7 – what do the authors mean by “off-DNA small molecule”? Do they mean synthetic?

AZ2429 was discovered using a DNA-encoded library screen against the PAR2-stabilised receptor and as such the molecule that bound to the receptor initially had a DNA tag attached which could then be used to identify what the chemical structure of the molecule was. Off-DNA small molecule refers to the small molecule without the DNA tag, as labeled AZ2429 in the text and illustrated in Figure 2. For clarity we updated the text by removing comment ‘off-DNA’.

10. Page 9 – there is a brief mention that interaction data between AZ3451 and 2f-LIGRL-NH2 or trypsin were fitted to an allosteric model to calculate cooperativity values. However, the model fits to the data should be shown, plus inclusion of a table showing all the parameters in the equation shown on page 17 (i.e. pK_A, pK_B, Logtau_A, Logtau_B, n, α , β , E_m, Basal), including any constraints that were applied during the analysis.

As suggested by the reviewer, we now include the graph in the SI (Supplementary Figure 5). We have also included the requested information in the Methods (Page 17, line 499–507).

“E_m is the maximum possible cellular response and held constant at 100, [A] and [B] are the concentrations of the orthosteric and allosteric ligands respectively. *n* represents the slope of the transducer function and held constant at 1. K_A and K_B are equilibrium dissociation constants for the orthosteric and allosteric ligands respectively, as determined with Eu-2f-LIGRLO-NH₂, pK_A of 2f-LIGRL-NH₂ = 6.1, pK_B of AZ3451 = 6.9. Parameters τ_A and τ_B denote the capacity, of orthosteric and allosteric ligands respectively, to exhibit agonism, log τ_A = 0.7 and log τ_B = 0 since AZ3451 showed no agonist activity.”

11. For the Schild analysis in Figure 4, it appears that the agonist E_{max} has been shared between curves. What was the rationale for this (ie. was an F test used to determine whether data were fitted best when the E_{max} was shared?). It is clear that at least in panel d, AZ3452 reduces the E_{max} of trypsin.

In Figure 4, E_{max} of the agonist was shared between data sets as we have used the Gaddum/Schild model (GraphPad Prism 8). The most rigorous method for quantifying agonist-antagonist interactions is to globally fit (i.e. share model parameters between data sets) the Gaddum/Schild model to all the agonist dose-response curves that were constructed in the absence and presence of different concentrations of antagonist (we added the reference <https://www.facm.ucl.ac.be/cooperation/Vietnam/WBI-Vietnam-October-2011/Modelling/RegressionBook.pdf>). Moreover, for this approach to work, the nonlinear

regression program must be able to perform global fitting to all data sets. Nonetheless, as suggested by the reviewer, we have also re-tested our data sets using a non-share model to generate the older, simpler, linear Schild regression plot. As shown in Figure R1 below, the calculated Schild slope remains consistent with Figure 4, further validating our findings that AZ8838 is a competitive antagonist while AZ3451 is a non-competitive antagonist.

Figure R1. Linear Schild regression plot for AZ8838 and AZ3451 measured by agonist (2f-LIGRL-NH₂ or trypsin) log dose ratios in the absence and presence of antagonist in CHO-hPAR2 cells. AZ8838 was a competitive antagonist against **a** 2f-LIGRL-NH₂ (slope = 1) and **b** trypsin (slope = 0.8). AZ3451 was a non-competitive antagonist against **c** 2f-LIGRL-NH₂ (slope = 0.4) and **d** trypsin (slope = 0.2).

F test was used to test whether a parameter differs from a hypothetical value and to statistically analyze our calculated Schild slope. In Figure 4, we used F test to determine if the calculated Schild slope is significantly different from a hypothetical value of 1. We have included more information on Gaddum/Schild analysis and F test in the methods (page 17 line 490 – 507) to better reflect our findings.

12. Page 10, discussion. The authors state that AZ3451 “was active in vivo in rats after subcutaneous injection but not after oral delivery.” Were the experiments following oral delivery performed in the current study but not shown or previously published?

Changed to “is active in vivo in rats after s.c. ... but not after oral delivery (not shown).”

13. Consistent with Tables 1 and 2, Table 3 should show pKd, not Kd.

As suggested by the reviewer and for consistency throughout the manuscript, we’ve now converted the remeasured Kd values in Table 3 to pKd values.

14. In Supplementary Figure 3e, AZ8838 in pERK1/2 assays does not appear to be approaching 100% inhibition, meaning it looks allosteric not competitive. If AZ8838 is truly competitive, it should show similar potency in all inhibition assays providing the same concentration of orthosteric agonist is present. Clearly either its potency or efficacy as an antagonist is reduced in pERK1/2 assays against trypsin. Any explanation?

We apologize for not making this clearer. AZ8838 was shown to bind in the 'putative' orthosteric site of PAR2, mimicking the interactions of the tethered ligand and demonstrated to be a competitive antagonist against the two agonists, GB110 and 2f-LIGRLO-NH₂ (Figure 1b). Since proteolytic activation of PAR2 is distinctively different from peptide activation, we suspect that different agonists can stabilize unique active conformations of PAR2 leading to biased signaling. This raises the possibility that the biased signaling induced by different agonists could influence the potency of AZ8838 against trypsin in pERK1/2. Due to compound solubility it was not possible to test AZ8838 at concentrations above 100 μM, however based on data shown we would anticipate that it would fully inhibit at higher concentrations.

Reviewer #3 (Remarks to the Author):

In this manuscript the authors expand on recent work from their laboratories to provide a handful of novel small molecule protease-activated receptor-2 (PAR2) binding compounds. They provide details on 6 antagonists (4 novel) from two distinct "series" and a novel antagonist. The authors do a nice job evaluating multiple in vitro signaling assays including: Ca²⁺ after activation of either human or rat receptor; inositol trisphosphate (IP1) as a Gq surrogate; pERK1/2 as a mitogen activated p kinase surrogate; and beta-arrestin; that, taken together, helps to clarify full agonist/antagonist potential. The authors additionally use a competitive binding assay, surface plasmon resonance (SPR) and single cycle kinetics followed by pharmacological analyses to evaluate compound interaction with PAR2. The authors finish their report with an in vivo response using a straight-forward paw oedema model for inflammation associated with disease. The authors argue that differences in pharmacology allow for a novel interpretation of at least two distinct binding sites for antagonists can allow for better bench tools in the laboratory and focus on the development of novel compounds for PAR2 antagonism. In all, the four new antagonists derived from previously published AZ8935 and AZ8838 as well as the novel agonist (AZ2429) are well characterized. There is some concern/confusion, however, as to the novelty of the report in that the two previously published compounds were already characterized as having different binding pockets (e.g., Cheng et al) and it has been proposed that these have different binding sites (what the authors are apparently referring to as "functional cross-talk"). In all, the new antagonist compounds represent important and relatively well-characterized additions for potential use in PAR2 drug discovery.

Major Concerns:

1) Abstract and throughout: it is unclear what is meant by "crosstalk" As far as can be determined from the data presented, the authors are describing two different antagonistic mechanisms/binding sites that have been previously proposed by this group (an orthosteric site and an allosteric site) with two of the compounds used in this manuscript (AZ8838 and A8935; Cheng et al., 2017). In the biological literature

crosstalk generally refers to the interaction of two distinct signaling pathways to cause a separate event. While the potential for signal transduction crosstalk is apparent with the functional bias that has been shown to occur with PAR2, the focus of the apparent cross talk described in this manuscript appears to be that “AZ8838 and AZ3451 have distinct modes of action” (Page 8, header). Unless this reviewer misinterpreted the data, the authors are discouraged in using “crosstalk” in describing the differences of orthosteric and allosteric inhibition. A proper “crosstalk” evaluation would include comparison of single compounds and combinatorial compounds that lead to a different signaling/physiological outcome. It would seem that a different description of the varied binding pockets might be in order to best describe the results presented.

We thank the reviewer for their suggestion and, having considered Reviewer 1 comment 3 as well, we have now replaced the title with ‘Protease-activated receptor-2 ligands reveal orthosteric and allosteric mechanisms of receptor inhibition’. We understand your point on the definition of crosstalk in biological literature and as such we have replaced ‘crosstalk’ throughout this manuscript.

2) AZ3451 and AZ8838 have already been reported in at least two separate articles from this group (Cheng, et al., 2017; Brown et al., 2018). These articles already describe the different proposed orthosteric and allosteric binding sites on PAR2. While the authors do clearly cite that these have been published, and there are new compounds presented, in the end it is the originally and previously published compounds that are used throughout as the “novel” compounds (e.g., all *in vivo* work is done with AZ3451 and AZ8838). Further, it is a bit of a stretch to call a group of two compounds (the “imidazoles” AZ8838 and AZ0107) a “series.” In summary there is concern about an overstatement of the novelty of the compounds presented.

Although AZ3451 and AZ8838 compounds and their binding sites within PAR2 in the solid state have been previously disclosed, this manuscript presents novel data which extensively characterises these compounds in solution across multiple signalling pathways and reveals the pharmacological mechanisms for inhibition by these two compounds. In addition, we report for the first time that these compounds have distinct mechanisms of inhibition and are both active *in vivo*. This is a complimentary manuscript to the Nature paper (Cheng et al. 2017), and we are delighted now to be able to report the full pharmacological characterisation of these PAR2 ligands. We are pleased that Reviewers 1 and 2 did not have concerns over novelty and thought ‘this work is a valuable extension of the innovative structural and med. chem. work from the AstraZeneca/Heptares/X-Chem team,’ and ‘represent novel tool compounds for studying PAR2’ respectively.

3) Binding was mostly assayed with a competitive binding assay but is listed as “Specific Binding” in Table 1 and throughout the manuscript (e.g., page 5, line 110). This should be corrected throughout the manuscript.

Thank you, this has now been corrected.

4) Activation of PAR2 varies throughout the manuscript. Early in the manuscript comparisons are made to the low potency SLIGVK-NH₂ peptide. However, as the manuscript moves on, other agonists are used, including GB110, 2f-LIGRL-NH₂, 2f-LIGRLO-NH₂. When trypsin is used, a concentration is given rather

than the Units of activity. It should be made clear why changes were made or why changes do not matter and how much trypsin is used for activation/antagonism experiments. This is especially important for readers that have not worked directly with PAR2, but also important for understanding potency/efficacy of antagonists across natural activators (i.e., trypsin units vs other serine proteases).

Additional information on why the use of different peptide agonists does not matter has been added on page 5.

Minor Concerns:

5) Page 3, line 47: "Protease-activated receptors (PAR)..." should show the abbreviation as "PARs" since throughout PAR refers to the singular.

Thank you, this has now been corrected.

6) Page 3, line 65: "PAR receptor" is redundant (i.e., protease-activated receptor receptor).

Thank you, this has now been corrected.

7) Page 5, line 121: "IP-One" should be IP-1, as used throughout.

Thank you, this has now been corrected.

8) Starting on Page 5, line 123 and throughout manuscript: nM values listed for EC50 do not match with values in Tables 1, 2. The use of two different numerations is confusing (pIC50 vs IC50), but somewhat understandable for ease of use in the Tables. However, conversions do not match. At the least, significant figures should allow for simple conversions. It is also unclear why the authors sometimes use 0.X microM vs x00 nM (e.g., page 7, lines 178 and 180).

As requested by both reviewers 2 and 3, the values within the text have now been changed such that both table and text have pIC50/pEC50/pKi with errors for consistency.

9) The authors seem to gloss over the different delivery of AZ8838 and AZ3451 in the evaluation of paw oedema (e.g., page 13, line 372; page 19, lines 545-546). Does this mean that AZ8838 does not work s.c? or AZ3451 not work p.o? The differences should be clearly stated.

Both compounds were found to be active after s.c. administration, but only one compound was active after oral delivery. This has been clarified in the discussion, page 10 lines 289 and 293.

Reviewers' comments:

Reviewer #1 (Remarks to the Author):

The modifications of the manuscript by Dekkar et al. look acceptable to me, with one minor exception detailed below. If Reviewer 2 is wanting confidence intervals (or something related) when comparing compound activities, this is overkill for in vitro SAR tables, especially for the data provided (with potencies differing by multiple log units). I will let Reviewer 2 (and potentially Reviewer 3) comment on the updated pharmacology data and analysis, but if the updates are reasonable this manuscript should be accepted without further delay.

Recommended minor edits:

1) The reductive amination reaction to give compound 6-2 must surely yield a mixture of diastereomers. What is the ratio? How do the authors know that they have the cis isomer? This appears to be a sloppy account of the purification, since presumably the other diastereomer was present but not described.

Reviewer #2 (Remarks to the Author):

While the authors have made some improvements to their manuscript, particularly with regards to statistical analysis of their data, there are a number of inconsistencies with their data that require convincing explanations before this manuscript can be considered for publication. My concerns are outlined below.

Major concerns:

1. Lines 115-117 "Throughout this manuscript, the reference agonists GB110, 2f-LIGRLO-NH2 and SLIGRL-NH2 are interchanged as they have each been shown to bind at a common or overlapping site within PAR2 and are reported to activate the receptor in a similar manner.". However, according to binding experiments, AZ3451 and AZ8833 bind with ~10 fold lower affinity when europium-labelled 2f-LIGRLO-NH2 was used as the labelled ligand, versus when 3H-acetylated-GB110 was used. Please can the authors provide an explanation for this to further justify that different agonists activate the receptor in a similar manner, as these results suggest evidence for different states stabilised by the different agonists.

2. According to the methods section, assays in Figure 1 showing antagonism of agonist responses were performed in the presence of an EC80 agonist concentrations. However, all responses appear to be performed in the presence of an Emax concentration of agonist. Please clarify.

3. Lines 162 – 164 "These mutagenesis studies suggest that AZ3451 binds to a high affinity site on PAR2, likely that observed in the crystal structure, while the biphasic behaviour is consistent with a second binding site with lower affinity for this ligand.". I think this needs further explanation. Why are the two sites only apparent when trypsin is used as the agonist?

4. Line 218 "AZ2429 association kinetics were around 100-fold slower, resulting in a 10-fold weaker affinity". Slow association kinetics do not necessarily result in a weaker affinity if dissociation kinetics are also slow. Please rephrase this sentence. Similarly, lin 222 – 224 "AZ3451 was able to bind to the receptor, albeit with a significant 3-fold reduction in affinity due to reduced association and dissociation rates.". It is the extent to which one of these rates is changed relative to the other that alters the affinity. E.g. if affinity is reduced, association must be reduced by a greater extent than dissociation is reduced.

5. Line 251 and figure 4c indicates that the binding cooperativity (α) between AZ3451 and 2F-LIGRL-NH2 is 0.01. However, in Figure 1b, AZ3451 completely inhibits the binding of 2f-LIGRLO-(dtpa)Eu, indicating that α is approaching 0 and is indistinguishable from a non-competitive interaction. Please explain the discrepancy between the two assays. Related to this point, on line 369 "explaining why AZ3451 induces partial inhibition of only the tethered ligand activated state." Yet Figure 4c and an $\alpha = 0.01$ clearly shows partial inhibition of the peptide, 2f-LIGRL-NH2. Please clarify and explain.

Minor comments:

6. Line 38 "allosteric antagonist" should be allosteric modulator.

7. Line 220 "To investigate the coupling between the antagonist". In light of the previous concerns of the reviewers regarding the use of "coupling" to describe the interaction between two allosteric sites, please rephrase.

8. Lines 220 and 226 – what is the "fragment binding site"? If the authors are referring to the AZ2429 binding site, I suggest referring to it as the AZ2429 binding site so as to not confuse the readers.

9. Line 292 "AZ3451, is a potent antagonist". The authors say that AZ3451 binds to a site distinct from the orthosteric binding site, therefore AZ3451 is a negative allosteric modulator, not an antagonist.

10. Line 429 Ca²⁺ methods, please state whether peak Ca²⁺ response or AUC is reported.

11. Please provide a description of beta in the operational model of allosterism starting on line 500, as well as beta values for the analysis.

12. Table 3 shows **** to denote statistically significant differences between the K_a , residence time and pK_d of AZ2429 in the presence of 50 μ M AZ8838 compared to AZ3451. However, the table denotes that there is no binding of AZ2429 under these conditions, therefore it is not possible to perform a statistical test on this data. Please remove the asterisks.

13. Line 304 and elsewhere, the StaR receptor is described as having mutations that "lock the receptor in an antagonist bound configuration". Can the authors clarify here whether the StaR is actually bound to an antagonist? If not, this should be described as a configuration akin to an inactive conformation.

Reviewers' comments:

Reviewer #1 (Remarks to the Author):

The modifications of the manuscript by Dekkar et al. look acceptable to me, with one minor exception detailed below. If Reviewer 2 is wanting confidence intervals (or something related) when comparing compound activities, this is overkill for in vitro SAR tables, especially for the data provided (with potencies differing by multiple log units). I will let Reviewer 2 (and potentially Reviewer 3) comment on the updated pharmacology data and analysis, but if the updates are reasonable this manuscript should be accepted without further delay.

Recommended minor edits:

1) The reductive amination reaction to give compound 6-2 must surely yield a mixture of diastereomers. What is the ratio? How do the authors know that they have the cis isomer? This appears to be a sloppy account of the purification, since presumably the other diastereomer was present but not described.

We agree with the reviewer that this was somewhat unexpected, but our analysis of the isolated product showed compound 6-2 to be a single isomer. Analysis of 6-2 on chiral HPLC with several methods only revealed a single stereoisomer. In terms of the assignment of the relative stereochemistry, we do say in the supplementary text that we are assigning the relative and absolute stereochemistry from the VCD experiment – but that we accept that it is tentative as reflected in current text that *'This result indicates that compound 6-2 has a S,R-configuration.'*

A sentence has been added in the supplementary materials on page 8 line 12 from top for clarity:

Analysis of 6-2 on chiral HPLC with several methods only revealed a single stereoisomer.

Reviewer #2 (Remarks to the Author):

While the authors have made some improvements to their manuscript, particularly with regards to statistical analysis of their data, there are a number of inconsistencies with their data that require convincing explanations before this manuscript can be considered for publication. My concerns are outlined below.

Major concerns:

1. Lines 115-117 "Throughout this manuscript, the reference agonists GB110, 2f-LIGRLO-NH2 and SLIGRL-NH2 are interchanged as they have each been shown to bind at a common or overlapping site within PAR2 and are reported to activate the receptor in a similar manner.". However, according to binding experiments, AZ3451 and AZ8833 bind with ~10 fold lower affinity when europium-labelled 2f-LIGRLO-NH2 was used as the labelled ligand, versus when 3H-acetylated-GB110 was used. Please can the authors provide an explanation for this to further justify that different agonists activate the receptor in a similar manner, as these results suggest evidence for different states stabilised by the different agonists.

In addition to the different ligands used for evaluation of compound binding affinity at PAR2; there are experimental difference in both assays. These differences include but not limited to:

- (a) Protein: The europium-labelled 2f-LIGRLO-NH₂ binding assay was carried out in whole cell (CHO-hPAR2), whereas the 3H-acetylated-GB110 assay was performed in membranes (HEK293-hPAR2)
- (b) Detection method: time-resolved fluorescence of europium versus radioactive decay.
- (c) Pre-incubation of ligands. For europium-tagged competition binding assay, CHO-hPAR2 cells are simultaneously exposed to both 2f-LIGRLO-(dpta)Eu and AZ3451 for 15 mins while in the tritium-labelled competition binding assay the label and AZ3451 were incubated together with the membranes for 2 hours.

The two antagonist series have a consistent rank order across both binding assays; the benzimidazoles have a higher affinity than the imidazoles. We suggest that the moderate difference in potency between assays is likely due to the experimental parameters and not that the ligands stabilise different 'active' receptor states. Based on the literature, our structural knowledge of the receptor and ligand interactions as well as the agonists pharmacological profiles, it is proposed that these receptor agonists bind in a common/overlapping binding site and therefore activate the receptor in a similar manner.

2. According to the methods section, assays in Figure 1 showing antagonism of agonist responses were performed in the presence of an EC₈₀ agonist concentrations. However, all responses appear to be performed in the presence of an E_{max} concentration of agonist. Please clarify.

Figure 1 was generated by evaluating the effect of the antagonists against EC₈₀ concentrations of SLIGRL-NH₂ (or trypsin Figure 1g). The EC₈₀ concentrations selected, as stated in the methods, were determined by first performing SLIGRL-NH₂ and Trypsin concentration response curves in the corresponding assay, this data is not included in the manuscript. The authors are unsure why the reviewer believes these assays were performed in the presence of E_{max} concentrations of agonist and seek clarification as to what evidence there is to suggest this.

3. Lines 162 – 164 “These mutagenesis studies suggest that AZ3451 binds to a high affinity site on PAR2, likely that observed in the crystal structure, while the biphasic behaviour is consistent with a second binding site with lower affinity for this ligand.”. I think this needs further explanation. Why are the two sites only apparent when trypsin is used as the agonist?

We agree with the reviewer that this is very intriguing pharmacology and is worthy of further discussion. As such, we included a paragraph (lines 350 – 377) in the discussion section of this manuscript to propose an explanation as to why this is only apparent when trypsin is used as the agonist. We have modified the statement at the end of this results section to simply state the findings ‘These mutagenesis studies suggest that AZ3451 binds with a high affinity to the site observed in the PAR2-AZ3451 crystal structure, but, intriguingly, has a second lower affinity binding site within the receptor which is only accessible when PAR2 is activated by the tethered ligand mechanism’.

Our proposed explanation of why the two sites are only apparent when trypsin is used as an agonist are later in the discussion section of the manuscript. Line 366 ‘Although a second binding site for AZ3451 was not previously found, it is perhaps because the StaR conformation does not allow binding to the second site, or the second site is only accessible upon proteolytic cleavage of the N-terminus.’ Alternatively, we propose that it only becomes apparent that AZ3451 has two binding sites when PAR2 is activated by trypsin as the high affinity binding site is disrupted by protease

activation of the receptor, and not during the peptide-induced active receptor state. Line 370 'It is possible that the movement of the tethered ligand into the binding site induces further conformational changes in the receptor, which might have a profound effect on the outside of the helical bundle where the high affinity binding site of AZ3451 is located, explaining why AZ3451 induces partial inhibition of only the tethered ligand-activated state.' The partial inhibition of trypsin signalling by AZ3451 means it is possible to see further inhibition by a second lower affinity site. In addition, we summarise the experimental evidence that suggests the biphasic response is not due to other factors and provide literature to support the proposed hypotheses.

4. Line 218 "AZ2429 association kinetics were around 100-fold slower, resulting in a 10-fold weaker affinity". Slow association kinetics do not necessarily result in a weaker affinity if dissociation kinetics are also slow. Please rephrase this sentence. Similarly, lin 222 – 224 "AZ3451 was able to bind to the receptor, albeit with a significant 3-fold reduction in affinity due to reduced association and dissociation rates.". It is the extent to which one of these rates is changed relative to the other that alters the affinity. E.g. if affinity is reduced, association must be reduced by a greater extent than dissociation is reduced.

These have been rewritten for clarity:

Line 219: AZ2429 association kinetics were around 100-fold slower, whilst the dissociation rate only decreased 5-fold, resulting in a 10-fold weaker affinity.

Line 224: AZ3451 was able to bind to the receptor, albeit with a significant 3-fold reduction in affinity due to a decreased association rate in combination with an increased dissociation rate.

5. Line 251 and figure 4c indicates that the binding cooperativity (α) between AZ3451 and 2f-LIGRL-NH₂ is 0.01. However, in Figure 1b, AZ3451 completely inhibits the binding of 2f-LIGRLO-(dtpa)Eu, indicating that α is approaching 0 and is indistinguishable from a non-competitive interaction. Please explain the discrepancy between the two assays. Related to this point, on line 369 "explaining why AZ3451 induces partial inhibition of only the tethered ligand activated state." Yet Figure 4c and an $\alpha = 0.01$ clearly shows partial inhibition of the peptide, 2f-LIGRL-NH₂. Please clarify and explain.

Despite similar ligands used in Figure 1b (europium-tagged competition binding assay) and Figure 4c (Ca²⁺ flux assay), there are key differences in both assays. These differences include but not limited to:

(a) Cells: CHO-transfected hPAR2 cells versus non-transfected HT-29 cells.

(b) Detection method: time-resolved fluorescence of europium versus fluorescent indicator of Ca²⁺ ions.

(c) Pre-incubation of ligands. For europium-tagged competition binding assay, CHO-hPAR2 cells are simultaneously exposed to both 2f-LIGRLO-(dtpa)Eu and AZ3451 while in Ca²⁺ flux assay, AZ3451 are pre-incubated for 60 min at 37°C prior to the additional of 2f-LIGRLO-NH₂.

The data should not be compared between the europium-tagged competition binding assay and the Ca²⁺ flux assay. The europium-tagged competition binding assay is designed to measure the affinity of a ligand, but not to characterise a ligand as an agonist or antagonist, while the Ca²⁺ flux assay is designed as one type of functional readout to characterise an agonist or antagonist. To highlight the difference between the europium-tagged competition binding assay and Ca²⁺ flux assay, we have

modified the methods (line 402) to include the details on the simultaneous exposure of both Eu-tagged 2f-LIGRLO-(dpta) and AZ3451 in the europium-tagged competition binding assay.

Minor comments:

6. Line 38 “allosteric antagonist” should be allosteric modulator.

This has been changed

7. Line 220 “To investigate the coupling between the antagonist”. In light of the previous concerns of the reviewers regarding the use of “coupling” to describe the interaction between two allosteric sites, please rephrase.

This has been changed to ‘To investigate the interplay between the different ligand binding sites of PAR2...’ to address both this comment and the following comment.

8. Lines 220 and 226 – what is the “fragment binding site”? If the authors are referring to the AZ2429 binding site, I suggest referring to it as the AZ2429 binding site so as to not confuse the readers.

This has been clarified in the manuscript.

9. Line 292 “AZ3451, is a potent antagonist”. The authors say that AZ3451 binds to a site distinct from the orthosteric binding site, therefore AZ3451 is a negative allosteric modulator, not an antagonist.

This has been changed in the manuscript.

10. Line 429 Ca²⁺ methods, please state whether peak Ca²⁺ response or AUC is reported.

The difference between the peak Ca²⁺ response and baseline response for Ca²⁺ flux is reported and this has been added into the methods section.

11. Please provide a description of beta in the operational model of allosterism starting on line 500, as well as beta values for the analysis.

We have now added the description and values for β in the methods for the operational model of allosterism.

12. Table 3 shows **** to denote statistically significant differences between the K_a, residence time and pK_d of AZ2429 in the presence of 50 μ M AZ8838 compared to AZ3451. However, the table denotes that there is no binding of AZ2429 under these conditions, therefore it is not possible to perform a statistical test on this data. Please remove the asterisks.

These asterisks have been removed from the manuscript – thank you

13. Line 304 and elsewhere, the StaR receptor is described as having mutations that “lock the receptor in an antagonist bound configuration”. Can the authors clarify here whether the StaR is actual bound to an antagonist? If not, this should be described as a configuration akin to an inactive conformation.

This has been clarified in the text. The StaR receptor does not have an antagonist bound but is an inactive-like configuration of the receptor.

REVIEWERS' COMMENTS:

Reviewer #1 (Remarks to the Author):

This reviewer finds that the new edits justify publication in Communications Biology without additional changes. The authors should be congratulated on their persistence and attention to detail.

Reviewer #2 (Remarks to the Author):

1. In response to the authors' rebuttal stating "Within the manuscript, AZ3451 is only referred to as an antagonist in summary statements encompassing both of the novel ligands presented that inhibit the activity of agonists at PAR2.", on page 5, line 112 onwards, the authors still state that "the antagonists could fully compete for binding against 3H-acetylated-GB110 in competition binding assays on HEKexp1293F membranes expressing hPAR2 (AZ3451 pKi = 7.9 ± 0.1 and AZ8838 pKi = 115.64 ± 0.1).". Similarly, in the methods to radiolabelled competition binding assays, AZ3451 is referred to as a competing ligand. Please amend the text accordingly to reflect the fact that allosteric modulators do not compete for binding against a radioligand (unless the radioligand is also allosteric).

2. The figure legend to supplementary Figure 6 says that "Curves were fitted according to the operational model of allosterism...". If indeed these are the curve fits to the operational model of allosterism, then the model has not been applied correctly for a number of reasons: (1) there is no concentration-dependent shift in the curves for 0.1 and 0.3 μM AZ3451; (2) there is a decrease in the bottom asymptote of the curves in panel B (the operational model of allosterism does not accommodate constitutive activity); (3) the 30 μM curve in panel A does not plateau at the same level as the curves for 1, 3 and 10 μM . If these curves are not the fit of the operational model of allosterism to the data, please either show the operational model of allosterism curve fits, or alternatively do not state in the figure legend that the curves were fitted according to the operational model of allosterism. Further, there is still no mention of the value of beta on page 9 line 254 onwards. Both alpha and beta should be reported in the results, not the methods. Further, these values should ideally be reported as log alpha and log beta with associated error.

3. Page 8, line 228. As mentioned in my earlier review, it is confusing to switch to calling these sites the "fragment binding site and the antagonist binding site on the outside of the helix bundle.". Please name these sites according to the ligand that binds to them.

Reviewer#2:

1. In response to the authors' rebuttal stating "Within the manuscript, AZ3451 is only referred to as an antagonist in summary statements encompassing both of the novel ligands presented that inhibit the activity of agonists at PAR2.", on page 5, line 112 onwards, the authors still state that "the antagonists could fully compete for binding against 3H-acetylated-GB110 in competition binding assays on HEKexpi293F membranes expressing hPAR2 (AZ3451 pKi = 7.9±0.1 and AZ8838 pKi =115 6.4±0.1).".

RESPONSE: This sentence has been changed to read 'In agreement, both series bound to HEKexpi293F membranes expressing hPAR2 (AZ3451 pKi = 7.9±0.1 and AZ8838 pKi = 6.4±0.1) in competition binding assays against 3H-acetylated-GB110'.

Similarly, in the methods to radiolabelled competition binding assays, AZ3451 is referred to as a competing ligand. Please amend the text accordingly to reflect the fact that allosteric modulators do not compete for binding against a radioligand (unless the radioligand is also allosteric).

RESPONSE:

We agree that allosteric modulators do not directly compete for binding against a radioligand (unless the radioligand is also allosteric), however they can of course indirectly compete by altering conformation at the orthosteric binding site.

We have now replaced "competing compounds" with "test compounds" in both the europium-tagged competition binding assay and radiolabelled competition binding assay method sections.

2. The figure legend to supplementary Figure 6 says that "Curves were fitted according to the operational model of allosterism...". If indeed these are the curve fits to the operational model of allosterism, then the model has not been applied correctly for a number of reasons:

(1) there is no concentration-dependent shift in the curves for 0.1 and 0.3 uM AZ3451;

RESPONSE: To demonstrate the concentration-dependent shifts for AZ3451, we now provide the EC₅₀ values for 2f-LIGRL-NH₂ and trypsin against AZ3451 at the different concentrations. (Supplementary Table 1).

(2) there is a decrease in the bottom asymptote of the curves in panel B (the operational model of allosterism does not accommodate constitutive activity);

RESPONSE: We have revised the curve fit of the operational model of allosterism for panel B.

(3) the 30 uM curve in panel A does not plateau at the same level as the curves for 1, 3 and 10 uM.

RESPONSE: We have revised the curve fit of the operational model of allosterism for panel A.

If these curves are not the fit of the operational model of allosterism to the data, please either show the operational model of allosterism curve fits, or alternatively do not state in the figure legend that the curves were fitted according to the operational model of allosterism.

RESPONSE: Additionally, we have also now included a second operational model of allosterism as well (Supplementary Fig 6c,d) to estimate the effect of an allosteric modulator in the presence of an agonist as described in Elhert¹ and cite a previous study stating that AZ3451 is a negative allosteric modulator². This alternative operational model calculates the relative activity and estimates γ value

for the agonist (2f-LIGRL-NH₂ or trypsin) in the presence of an allosteric modulator (AZ3451). This addition supports the conclusion that AZ3451 is a negative allosteric modulator, as pointed out now by two different operational models of allosterism.

Further, there is still no mention of the value of beta on page 9 line 254 onwards. Both alpha and beta should be reported in the results, not the methods. Further, these values should ideally be reported as log alpha and log beta with associated error.

RESPONSE: We have now presented the log α and log β values with errors in the results as requested (page 9, line 269).

To reflect the above changes, we have:

1) added a section in the results (page 9, line 270) “A second model of allosterism¹ based on Ca²⁺ mobilisation allowed calculation of relative activity and estimation of the γ values for 2f-LIGRL-NH₂ and trypsin (Supplementary Table 1). The relative activity estimates the value of γ to be 0.002 and 0.09 for 2f-LIGRL-NH₂ and trypsin respectively, indicating that AZ3451 caused a decrease in both affinity and efficacy (Supplementary Figure 6c,d); consistent with a previous study showing that AZ3451 is a negative allosteric modulator of PAR2².”

2) revised the methods (page 17, line 550) to include a second operational model “Allosterism for AZ3451 in the calcium flux assay was determined and analysed as described¹. The relative activity was calculated using $(E_{\max}EC_{50}')/(E_{\max}'EC_{50})$ with the Hill slope held constant at 1. EC₅₀' and E_{max}' denote the EC₅₀ and E_{max} in the absence of allosteric modulator, while EC₅₀ and E_{max} denote those measured in the presence of allosteric modulator. When the Hill slope = 1, the relative activity plot approaches an asymptote corresponding to the estimate of the γ value.”

Supplementary Figure 6. Negative allosteric modulation of Ca²⁺ mobilization by AZ3451 in CHO-hPAR2 cells. Concentration response curves for (a) 2f-LIGRL-NH₂ or (b) trypsin were measured in the presence of increasing concentrations of AZ2341. Curves were fitted according to an operational model of allosterism and AZ3451 exhibits classical rightward and downward curve shifts of a negative allosteric modulator ($\alpha < 1$) against both 2f-LIGRL-NH₂ and trypsin. Data presented as mean

± s.e.m of n ≥ 3 independent experiments. Relative activity of (c) 2f-LIGRL-NH₂ or (d) trypsin calculated from Supplementary Table 1 using Ca²⁺ mobilisation in the presence of increasing concentrations of AZ2341. Curves were fitted with a 4-parameter fit and data presented as mean of n ≥ 3 independent experiments.

Supplementary Table 1. Calculated values of E_{max}, EC₅₀ and relative activity for 2f-LIGRL-NH₂ and trypsin in the presence of AZ3451 in CHO-hPAR2 cells.

Agonist	AZ3451 (μM)	E _{max}	EC ₅₀ (μM)	Relative activity
2f-LIGRL-NH ₂	30	55	94.6	0.002
	10	55	28.4	0.005
	3	53	15.4	0.009
	1	58	10.1	0.01
	0.3	72	4.8	0.04
	0.1	72	2.6	0.07
	0	100	0.25	1
Agonist	AZ3451 (μM)	E _{max}	EC ₅₀ (μM)	Relative activity
Trypsin	30	61	0.038	0.09
	10	74	0.035	0.12
	3	89	0.030	0.17
	1	90	0.027	0.20
	0.3	90	0.019	0.28
	0.1	96	0.013	0.43
	0	102	0.006	1

3. Page 8, line 228. As mentioned in my earlier review, it is confusing to switch to calling these sites the “fragment binding site and the antagonist binding site on the outside of the helix bundle.”. Please name these sites according to the ligand that binds to them.

RESPONSE: This has been changed in the manuscript.

References

- 1 Ehlert, F. J. Analysis of allosterism in functional assays. *J Pharmacol Exp Ther* **315**, 740-754, doi:10.1124/jpet.105.090886 (2005).
- 2 Kennedy, A. J. *et al.* Structural Characterization of Agonist Binding to Protease-Activated Receptor 2 through Mutagenesis and Computational Modeling. *ACS Pharmacol Transl Sci* **1**, 119-133, doi:10.1021/acspsci.8b00019 (2018).